



# Flood forecasting with machine learning models in an operational framework

Sella Nevo[1], Efrat Morin[2], Adi Gerzi Rosenthal[1], Asher Metzger[1], Chen Barshai[1], Dana Weitzner[1], Dafi Voloshin[1], Frederik Kratzert[1], Gal Elidan[1,2], Gideon Dror[1], Gregory Begelman[1], Grey Nearing[1], Guy Shalev[1], Hila Noga[1], Ira Shavitt[1], Liora Yuklea[1], Moriah Royz[1], Niv Giladi[1], Nofar Peled Levi[1], Ofir Reich[1], Oren Gilon[1], Ronnie Maor[1], Shahar Timnat[1], Tal Shechter[1], Vladimir Anisimov[1], Yotam Gigi[1], Yuval Levin[1], Zach Moshe[1], Zvika Ben-Haim[1], Avinatan Hassidim[1], Yossi Matias[1]

[1]Google Research, Yigal Alon 96, Tel-Aviv 6789141, Israel
[2]Hebrew University of Jerusalem, Safra Campus, Jerusalem 91904, Israel

10    *Correspondence to*: Efrat Morin (efrat.morin@mail.huji.ac.il)

**Abstract.** Google's operational flood forecasting system was developed to provide accurate real-time flood warnings to agencies and the public, with a focus on riverine floods in large, gauged rivers. It became operational in 2018 and has since expanded geographically. This forecasting system consists of four subsystems: data validation, stage forecasting, inundation modeling, and alert distribution. Machine learning is used for two of the subsystems. Stage forecasting is modeled with the Long Short-Term Memory (LSTM) networks and the Linear models. Flood inundation is computed with the Thresholding and the Manifold models, where the former computes inundation extent and the latter computes both inundation extent and depth. The Manifold model, presented here for the first time, provides a machine-learning alternative to hydraulic modeling of flood inundation. When evaluated on historical data, all models achieve sufficiently high-performance metrics for operational use. The LSTM showed higher skills than the Linear model, while the Thresholding and Manifold models achieved similar performance metrics for modeling inundation extent. During the 2021 monsoon season, the flood warning system was operational in India and Bangladesh, covering flood-prone regions around rivers with a total area of 287,000 km$^2$, home to more than 350M people. More than 100M flood alerts were sent to affected populations, to relevant authorities, and to emergency organizations. Current and future work on the system includes extending coverage to additional flood-prone locations, as well as improving modeling capabilities and accuracy.



## 1. Introduction

Floods are a major natural threat to populations worldwide, causing thousands of fatalities and resulting in large economic damages annually. Jonkman (2005) analyzed a natural disaster database (EM-DAT, 2021) and found that over a 27 year period more than 175,000 people were killed and close to 2.2 billion were affected by floods worldwide. These numbers are likely an underestimation due to unreported events. Furthermore, UNISDR (2015) reported flooding to be the most frequent weather-related natural disaster, affecting the largest number of people globally, and with annual economic damage of more than $30 billion. Population growth, urbanization, and the changing climate have led to an increase in these numbers in recent decades - a trend that is likely to continue (Blöschl et al., 2019). Flooding vulnerability is especially significant in low-income and middle-income countries where adequate flood mitigation measures are often lacking or of limited quality and floodplains are often heavily populated (Alfieri et al., 2018). In such regions, operational flood warning systems are key to saving lives and reducing risks and damages (Hallegatte, 2012; World Meteorological Association, 2013).

Operational flood warning systems vary in their structure, data sources, and model types, depending on their specific target region, size of basins, available data and resources, and the system development approach (e.g, Emerton et al., 2016; Georgakakos, 2017; Krajewski et al., 2017; Shrestha et al., 2015; Werner et al., 2009; among many others). Many of the systems include real-time or forecast weather forcing data as an input into hydrological models, which in turn compute runoff and route flow through the river network, outputting streamflow at locations of interest. Flood warning systems often include an inundation modeling component (Teng et al., 2017) that translates the forecasted streamflow into a mapping of flood extent and depth (e.g., Bhatt et al., 2017; de Almeida et al., 2012). Inundation patterns in large rivers are often complex and depend on the river's morphology (e.g., meandering or braided rivers) and floodplain properties, such as the topography and land cover. They are also influenced by human actions, for example, structures that alter the natural flow of the water over the floodplain.

While flood warning systems are deployed in operational frameworks in different regions across the world, there is a lack of evaluation studies of such systems, presumably resulting from the difficulty in accessing ground truth data and the lower priority of the evaluation task in an operational environment (exceptions can be found in Welles and Sorooshian, 2009; Zalenski et al., 2017). It is important, however, to encourage agencies and companies responsible for flood warning operations to invest effort in evaluating and reporting performances, not only for assessing the forecast and warning reliability but also for benchmarking and developing the common scientific knowledge that can lead to further improvement and innovation. In the current study, we hope to contribute to this scientific effort.

Two main modeling components are important for flood warning systems, these are (i) a streamflow forecast model and (ii) an inundation model. Recent advances in the accuracy of data-driven and machine learning (ML) methods have affected a large variety of real-life applications (e.g., He et al., 2016; Devlin et al., 2019), and encourage the use of such methodologies as core drivers for those two modeling components. Earlier studies showed the potential of such models (e.g., Hsu et al., 1995; Tiwari and Chatterjee, 2010). More recent studies have found ML, and especially deep learning methods, to be a promising



approach for streamflow modeling, providing improvements over prominent conceptual models in prediction accuracy,
scalability, and regional generalization (e.g., Mosavi et al., 2018). Kratzert et al. (2019b) showed improved predictions relative
to conceptual models by using Long Short-Term Memory (LSTM) deep neural networks in more than 500 basins in the US.
In this case, regionally calibrated ML models outperformed not only regionally calibrated conceptual models but also
conceptual models that were calibrated for each basin separately. Kratzert et al., (2019a) found that even in basins that
contributed no training data (i.e., effectively ungauged basins), the LSTM models performed better than conceptual models
that were calibrated to long data records in every basin (i.e., in gauged basins). Xiang and Demir (2020) demonstrated high
skill in streamflow forecasting using a deep recurrent neural network for 125 stream gauges in Iowa, US.

ML methods have been shown to be promising for flood inundation modeling as well, providing a plausible alternative to
physically-based hydraulic models, that are both highly computationally demanding and challenging to use in operational flood
forecasting systems (Kabir et al., 2020). For example, artificial neural networks were utilized by Chang et al. (2018) and Chu
et al. (2020) to provide forecasts of flood inundation maps for rivers in Malaysia and Australia, respectively; and, Kabir et al.
(2020) utilized convolutional neural networks to predict with high accuracy spatially distributed water depths for large flood
events in an urban area in the UK.

While previous studies provided encouraging results, it is rare to find actual operational systems with ML models as their core
components that are capable of computing timely and accurate flood warnings. In 2017, Google launched the Flood Forecasting
Initiative, which aims to utilize Google's data, computational resources, and ML expertise to reduce flood-related fatalities
and harms. Since then, these efforts have been progressing in scale and in accuracy, with the aim of eventually providing
highly accurate flood forecasts globally. The first version of Google's flood warning system was operational from 2018,
initially at a limited scale in India. Following further development of ML modeling components and a substantial improvement
in coverage and performance, the system provided forecasts for the majority of India and Bangladesh in the monsoon seasons
of 2021.

This paper is aimed at two goals: (i) to present Google's end-to-end operational flood warning system and its current
deployment in India and Bangladesh for the monsoon season 2021 (Section 2); (ii) to evaluate ML models for river stage
forecasting and for inundation used in this system (Section 3); and, (iii) to present a new ML methodology for modeling flood
inundation extent and depth (the Manifold model, Section 2.3b). Since the results describe an operational deployment, we hope
to contribute to reducing the aforementioned scarcity of information on operational performance by assessing the utility of the
different ML models presented here in an operational framework.

## 2.   End-to-end real-time operational flood warning system

Google's end-to-end flood warning system is designed to obtain real-time data from different sources and to forecast future
river stages and high-resolution flood inundation at locations along the river network. These forecasts are disseminated in the
form of flood alerts to relevant agencies, as well as being sent out directly to the public. In its present form, the system is





designed for deployment in large gauged rivers, implying stream gauge data availability, relatively slow response, and potentially complex inundation areas during floods.

Stream gauges in locations of interest are defined as target gauges. Each target gauge has a pre-defined warning stage threshold, above which an alert should be issued, and a pre-defined *maximal lead time*, which is the maximal time in the future the

forecast can be given for. In addition, an area of interest (AOI) is defined for each target gauge, representing the surrounding region around the river and the gauge where the inundation model is applied.

This system is currently based on river stage data rather than discharge, which is more commonly used in physics-based and conceptual hydrological models. This is due to the wide abundance of stage data (historical and real time), while discharge data are more scarcely available and suffer from inaccuracies related to uncertainties in stage-discharge relationships (Ocio et

al., 2017; McMillan & Westerberg, 2015). Whereas it is difficult to use stage data to calibrate conceptual hydrology models (Jian et al., 2017), deep learning models do not have this problem and can be trained directly on stage data.

The general structure of the real-time flood warning system is outlined in Figure 1. Since Google's system is designed to be implemented in many countries and eventually globally, it is described first in general terms in Sections 2.1-2.4, followed by Section 2.5 that describes the details of the deployment in the 2021 monsoon season.

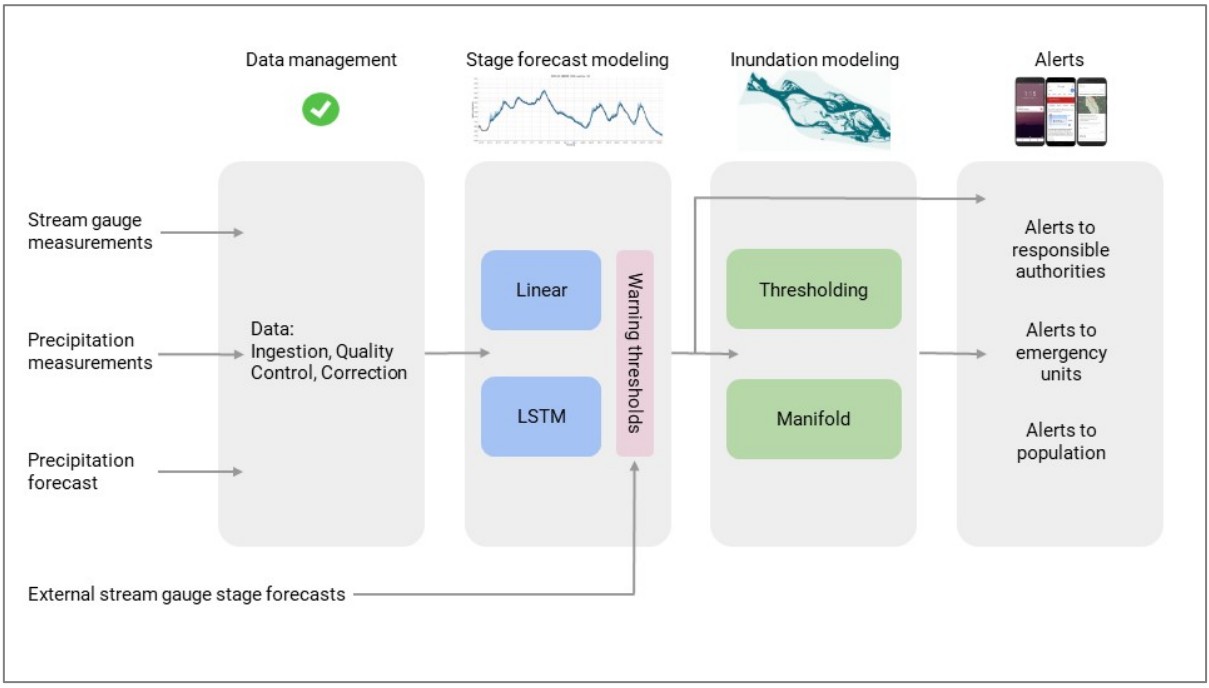

**Figure 1.** The operational flood warning system scheme. The first step is the ingestion, quality control and correction of stream stage measurements, precipitation measurements and precipitation forecast data. The validated data serve as an input to the stage forecast models (Linear and LSTM models) in the second step. An inundation model (Thresholding or Manifold) is applied in the third step for forecasted stages that pass the warning threshold. The final step is the dissemination of alerts to

relevant agencies, including emergency forces, and to the public.



## 2.1. Data management

In the first step, the following real-time data are ingested, quality controlled, corrected, and pre-processed:

Stream gauge measurements of water stage (obtained from the data provider, e.g., national water authorities): Stage data are

acquired in near-real-time for the target gauges and their upstream gauges, if available. In some countries, the stage is measured manually and prone to many kinds of human mistakes; even automated telemetry gauges frequently produce erroneous data. Therefore, all near-real-time stage data go through a series of validation and correction procedures as follows: (i) typical manual errors are identified (e.g., a decimal point in the wrong place) and corrected; (ii) stage record statistics are used to identify unreasonable stage data or stage differences from the previous readings (the suspicious data are either removed or

retained, based on a set of ad-hoc criteria); and, (iii) short periods of missing data are filled by linear interpolation. The temporal resolution of near-real-time stage data may vary among providers. The currently deployed systems in India and Bangladesh both use hourly stage data.

Satellite-derived precipitation data from the Integrated Multi-satellitE Retrievals for GPM (IMERG, Early Run) (Huffman et al., 2014): IMERG Early Run data is available with a latency of 12 hours, quasi-global coverage and a resolution of 0.1° in

space and 30-min in time. Data are quality controlled and corrected by removing negative and missing data values and by applying an upper rain intensity threshold (set as 200 mm/h for the current deployment). The data are spatially averaged over the drainage area of each target gauge, and integrated in time to match the temporal resolution of near-real-time stage data (hourly in the current deployment).

## 2.2. Stage forecast modeling

The stage forecast models are responsible for computing forecasted river stage data at target gauges. The system includes two types of models: 1) A model based on multiple linear regression, and, 2) a model based on a Long Short-Term Memory (LSTM) network, a type of recurrent neural network which is structured to learn long-term input-output dependencies. Linear models have been used in operational systems to simulate relationships between upstream and downstream river stages (In India for

example, World Bank, 2015), whereas the LSTM has been shown in recent years to improve hydrological simulations relative to conceptual models (e.g., Kratzert et al., 2018; Hu et al., 2019; Feng et al., 2020; Xiang et al., 2020).

(a) Linear model (Figure 2a): The model input includes past river stages from the target gauge and its upstream gauges (typically 2-5 gauges) for each time step. The output is a future river stage at the target gauge for a given lead time. A multiple linear regression model is trained with historical records of the above inputs and outputs. The model is optimized using the

mean square error (MSE) loss function with L2 regularization (i.e., a squared sum of coefficients is added as a penalty term). Linear models are trained separately for each target gauge, with a separate linear model trained for all lead times up to the gauge's *maximal lead time* (for example, a target gauge with a selected *maximal lead time* of 24 hours and hourly resolution implies 24 trained Linear models).





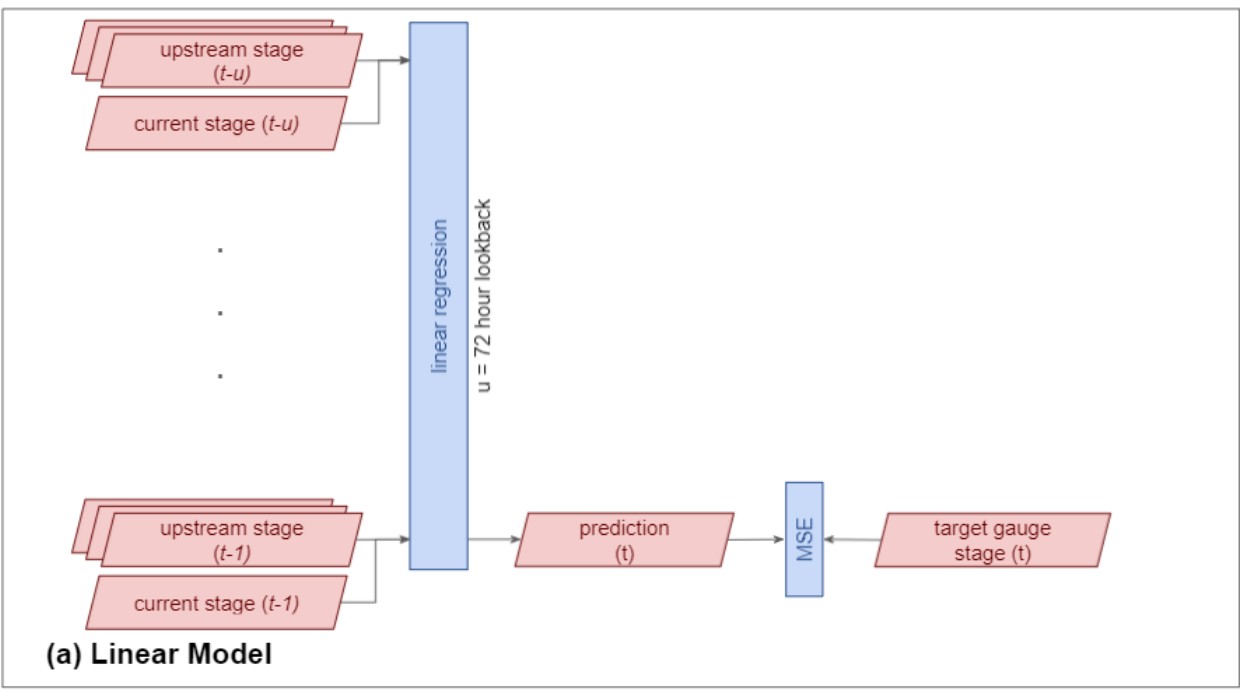

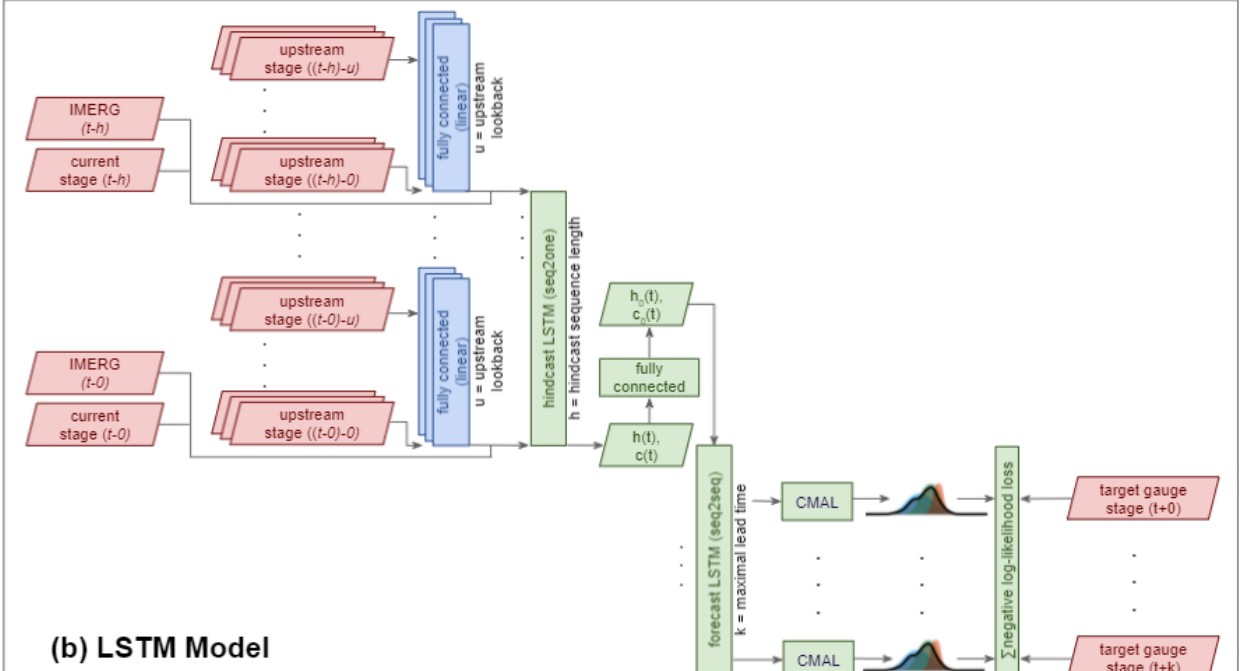

**Figure 2.** Schemes of the Linear (a) and LSTM (b) stage forecast models.





(b) Long Short-Term Memory network (LSTM) (Figure 2b): The LSTM model consists of two LSTMs: a sequence-to-one hindcast model and a sequence-to-sequence forecast model, where the output of the hindcast model is used as the initial state

of the forecast model. The hindcast model processes data from past days sequentially, taking the following variables as inputs at each timestep: (i) IMERG precipitation, (ii) near-real-time stage, and (iii) a linear combination of stage measurements from upstream gauges over some upstream "lookback" period (typically a few days). The linear combination of stage measurements is produced by a separate linear layer with gauge-specific weights that combines a (variable) number of upstream inputs per gauge into five features that are fed as inputs into the hindcast LSTM. The hindcast LSTM runs until the current time (defined

to be the time of the last available measurement). The final cell state and hidden state ($c(t)$ and $h(t)$ in Figure 2b) of the hindcast LSTM are passed through a fully connected layer and the output of this "state handoff" layer is used as the initial cell state and hidden state ($c_0(t)$ and $h_0(t)$ in Figure 2b) of the forecast LSTM. The forecast LSTM advances one step for every lead time, producing the relevant forecasts for all lead times up to the maximal one. The basic principle of using a state handoff between two LSTMs was adopted from Gauch et al. (2021). In the original use, the state handoff allowed modeling at multiple

timescales, while here we use this structure to differentiate between the different inputs that are available in real-time during hindcast and forecast (specifically, the availability of precipitation measurements).

All weights of the LSTMs (hindcast and forecast) are shared between all target gauges (i.e., they are regionally calibrated). The only gauge-specific weights are those of the linear upstream combiner layer which allows the architecture to accommodate varying numbers of upstream gauges, travel times, sizes of tributaries, etc.

The system estimates the uncertainty of the water stage following the approach detailed in Klotz et al. (2021). The time-dependent distribution over the predicted stage is modeled using a (countable) mixture of asymmetric Laplacians (CMAL). The parameters of this distribution are generated by feeding the hidden state of the forecast LSTM into a dedicated head layer for each forecasted time step. At each time step, the loss is calculated as the negative log-likelihood of the observed stages given the LSTM forecasts using the next *maximal lead time* values. It is important to note that the likelihood-based loss

function is calculated only over the outputs of the forecast LSTM, and the hindcast LSTM is used only to initialize the forecast state. Since training is shared for all target gauges, the *maximal lead time* in the training phase is taken as the maximum of the gauge-specific values.

Training and validation: The stage forecast models are trained and validated with historical data using a cross-validation

scheme, in which each fold uses one year's worth of data for validation and the rest for training (i.e., 1-year leave-out). For operational use, the models were retrained on the full data set to produce the best real-time forecasts possible.

Exceedance of warning threshold and output to the inundation model: For each target gauge, if the maximal forecasted river stage between the forecast's "current time" and the gauge's *maximal lead time* is above the predefined gauge-specific warning threshold, an alert is issued, and this maximal stage is used for inundation mapping (Section 2.3).

External forecasts: In addition to stage forecasts derived by the models described above, the flood warning system allows for using forecasts from external sources, such as models operated by the relevant national hydrology or hydrometeorology





authorities. External forecasts are also compared with gauge-specific warning thresholds and, if exceeded, are used for inundation mapping (Figure 1).

## 2.3. Inundation modeling

The inundation models are responsible for translating forecasted river stages into flood extent maps that delineate the projected flooded regions in areas of interest around gauges (i.e., AOIs). An additional potential output is the inundation depth at each flooded pixel in the AOI. In large rivers, the water stage varies relatively slowly with time due to the large volumes involved; therefore, the flood extent is computed based on the forecasted stage for a given time, ignoring the river stages at previous

time points (i.e., assuming the river flow is in a steady state).

Two ML models are currently utilized: 1) the Thresholding model, and, 2) the Manifold model. The Thresholding model computes a flood inundation map; it does not require a digital elevation model (DEM) and is easier to implement and deploy at scale, but it forecasts only flood extent (and not depth). The Manifold model can be implemented where a DEM is available and it computes both a flood inundation map and an inundation depth map. A third, physically-based Hydraulic model, which

also requires a DEM, is presented here for reference; it was used in the operational framework in previous seasons, but ultimately this model was discarded because it was found to be both less accurate and less scalable than the ML models (see Section 3.2), while its computational cost was considerably higher (Ben-Haim et al., 2019).

(a) Thresholding model (Figure 3a): The model assumes that each pixel in the AOI becomes inundated when the target gauge exceeds a (pixel-specific) threshold water stage. These thresholds are learned from the series of historic stage data at the target

gauge and the corresponding state of the pixel (dry/wet) during these events (Figure 3a). Each pixel in the inundation map is treated as a separate classification task, predicting whether the pixel will be inundated or not. We refer to the "wet" class as the positive class.

The algorithm described below identifies pixel-specific thresholds and is aimed at maximizing some F-score using an optimized global parameter called *minimal ratio*. An iterative process is applied to each pixel. In each iteration, we find the

threshold that maximizes the ratio of true wet events (where the water stage at the gauge is above the threshold and the pixel was wet) to false wet events (where the water stage at the gauge is above the threshold and the pixel is dry). The threshold that maximizes this ratio is the most cost-effective threshold in the sense that it provides the most true wets per false wet instance. At the first iteration all training events are considered; then, after each selection of a threshold and its respective true-false ratio, events with stage measurements above the threshold are discarded and a new iteration starts with the remaining events.

If the new true-false ratio calculated is lower than the *minimal ratio* parameter value, the process stops and the final threshold for the pixel is the one found in the previous iteration. It can be shown that for every *minimal ratio* parameter value, no other set of pixel-specific thresholds achieves simultaneously better precision (i.e., fraction of all flooded pixels that are predicted as being flooded) and recall (i.e., fraction of all pixels that are predicted to be flooded and are really flooded); implying it is





Pareto optimal. Therefore, for any F-score there exists some value of the *minimal threshold* parameter which finds the
thresholds that optimize this F-score.

In cases where the river stage input is higher than all past stage data, the Thresholding model's output inundation map is initialized from the most severe inundation extent seen in the historical events and expanded in all directions. The expansion distance is a linear function of the difference between the forecasted stage and the stage of the highest historical event. This Thresholding model requires almost no site-specific data like DEMs, and no manual work, making it appealing for large scale
deployment across many AOIs in a short amount of time.

(b) Manifold model (Figure 3b): The Manifold inundation model, presented here for the first time, provides a machine-learning alternative to hydraulic models, by computing physically reasonable flood inundation. Its inputs are a DEM for the AOI and a target water stage. It outputs both the flood inundation extent and the inundation depth at each AOI pixel. The model is divided into two major parts, as can be seen below.

(b.1) Flood extent to water height algorithm

The flood extent to water height algorithm converts a DEM and an inundation extent map (i.e., wet/dry state for each pixel) into a water height map, which is a per-pixel water height in meters above sea level. The algorithm tries to find a *physically reasonable* water height map that best matches the input inundation map, where the physically reasonable requirement is defined as: (1) the water height surface must be smooth, i.e. we aim to find a water height map that does not change significantly
between neighboring pixels; and, (2) the water height surface should not have a minimum or a maximum at the interior of flooded regions. This optimization problem is not differentiable, and thus cannot be easily solved directly. Instead, the following heuristic can be shown to produce an optimal solution to the above optimization problem. The algorithm identifies the boundaries of the inundated areas of the input inundation map (OpenCV findContours, 2021). The water height at these boundaries is extracted from the corresponding DEM. In between these boundaries, the algorithm uses the Laplace differential
equation to interpolate the water heights. The water height map is defined as a low-resolution image, where every pixel is set to be of 32x32 DEM pixels. This assures that the output map is smooth and does not contain high frequency changes, while also reducing the computational complexity of the process. In addition, outlier DEM pixels, which are pixels that cause high Laplace tension, are removed to assure that the overall function is smooth.

(b.2) Gauge stage to flood depth algorithm

When inferring inundation depth for a real-time gauge water stage forecast, we do not have access to the full current flood extent (as the extent-to-height algorithm above assumes). To be able to provide depth in this more challenging setting, we first perform some precomputation. We first apply the *Thresholding model* described above to all past events in our training data, producing an inundation extent map for each gauge stage measurement. We then apply the *flood extent to water height algorithm* described above to produce a water height map for each past event. In real-time, when we receive an input water
stage at the target gauge, the model simply performs per-pixel piecewise-linear interpolation between the water height maps of the training dataset to generate a new water height map corresponding to this input stage. The resulting water height map and the DEM are then used to generate: (1) an updated inundation extent map, by assigning a dry state to a pixel if its water



height is lower than the DEM and a wet state otherwise; and, (2) an inundation depth map, as the difference between the water height and the DEM height for wet pixels. The use of the Thresholding model ensures that the input inundation maps (as opposed to directly using satellite-based imagery of actual historical flood extent described below) ensure that higher gauge stages always yield larger inundation extents, and thus removes unnecessary noise. When the model infers an inundation depth map for a gauge's water stage higher than all the events in the training set, it extrapolates the water height map by adding the gauge level difference to every pixel in the highest water height map computed from observed gauge stages, and uses the extrapolated water height and DEM to compute the water depth map.

(c) Hydraulic model (Ben-Haim et al., 2019) (Figure 3c): The hydraulic model is a physics-based model based on numerical solutions of the St. Venant equations (de Almeida et al., 2012). In this implementation we assume that the river stage changes slowly, so that the flood extent at any given moment is approximated by a steady-state simulation. This is a good approximation for large rivers, which are the current focus of the flood warning system, where water stage changes relatively slowly due to the large amounts of water involved.

Such physics-based models are commonly used with incoming discharge data as one of the boundary conditions. However, in our systems the data available in real-time is the water stage. To support this the model is simulated off-line with numerous discharge values at the upstream boundaries. Other model parameters (e.g., roughness per AOI's pixel) and downstream slope boundary conditions are fixed to physical constants. Each run of the model yields a simulated inundation map for the AOI and a water stage at the target gauge. These results are saved in a hash table for quick retrieval. Upon receiving a real-time gauge stage forecast, the simulation with the closest gauge water stage is found and the corresponding inundation map is used as the model output.

The hydraulic model requires a DEM for the entire AOI, including the river bathymetry that is often not included in DEMs based on optical imagery (as well as other sources that cannot penetrate water). It is therefore estimated by fitting 3-equal-side trapezoids to the river cross sections with slopes that are computed from the river banks.

Training and validation: The inundation models are trained and validated based on historical flood events, where flood inundation extent maps from satellite data, along with the corresponding gauge water stage measurements, are available. Similar to the stage forecast models, a 1-year leave out cross validation scheme is used for training and validation. For operational use, the models are retrained with all historical data. It should be noted that, contrary to flood inundation extent, there is no ground truth data for flood inundation depth. The Manifold model is, therefore, trained and validated only on the inundation extent. However, since the Manifold model is constrained to only produce physically reasonable water height maps, accurate inundation extent metrics on the test dataset imply reasonably reliable inundation depth results. More work on validating inundation depth accuracy at scale is needed.

Satellite-based flood inundation maps: The synthetic aperture radar ground range detected (SAR GRD) data from the Sentinel-1 satellite constellation are used to determine flood inundation maps at known timepoints and locations (Schumann and Moller, 2015). At any AOI, a SAR image is available once every several days, from which an inundation map was inferred using a


binary classifier (Torres, 2012). Every pixel within a SAR image is classified as wet/dry via a Gaussian-mixture based classification algorithm. In order to calibrate and evaluate the classification algorithm, we have collected a dataset of Sentinel 2 multispectral images of flood events that coincide with the SAR image dates and locations. Reference Sentinel-2 flood maps

were created by calculating per-pixel Normalized Difference Water Index (NDWI=(B3-B8)/(B3 + B8), B3 and B8 are green and near infrared bands, respectively) and applied a threshold of 0 (McFeeters, 2013).

DEMs for target gauge AOIs: The publicly available global DEMs (e.g., NASA SRTM and MERIT) lack the required spatial accuracy and resolution for detailed flood inundation simulation. Furthermore, they are based on data from over a decade ago, thus failing to capture the frequent topography changes caused by past floods. Consequently, higher-resolution, up-to-date

DEMs are constructed for each AOI from high resolution satellite optical imagery data in a process that is based on stereographic imaging (Ben-Haim et al., 2019). To keep the DEM up to date, the model is retrained annually based on fresh imagery in locations where flooding causes frequent topography changes.

Model selection for target gauges: The selection of which model to use for each target gauge is based on DEM availability for the AOI and on the performance when trained and tested with historical data. When possible, the Manifold model is preferred

as it provides forecasts of inundated water depths in addition to the inundation map.

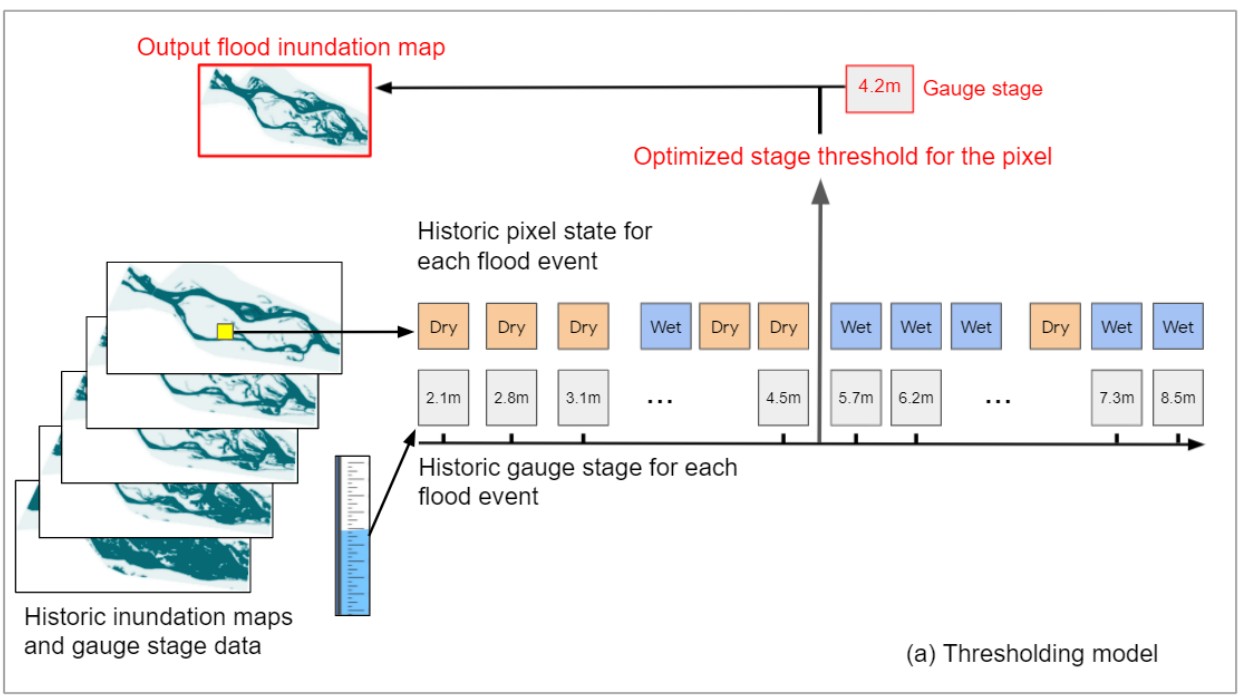



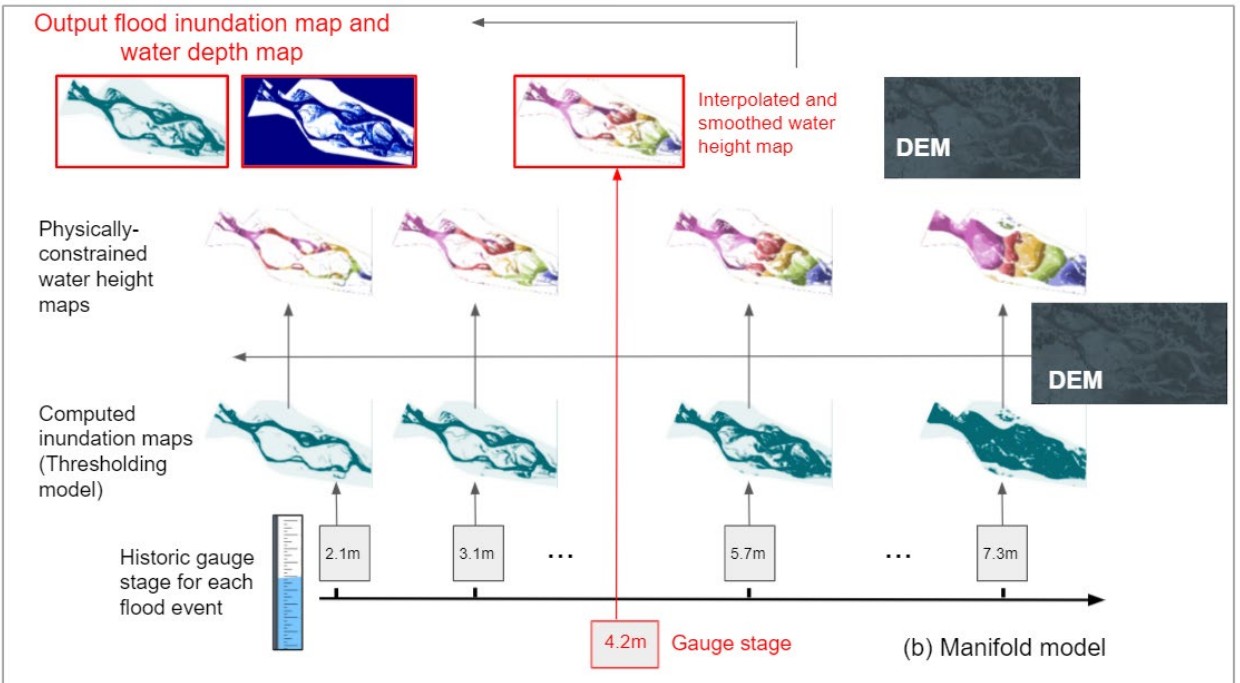

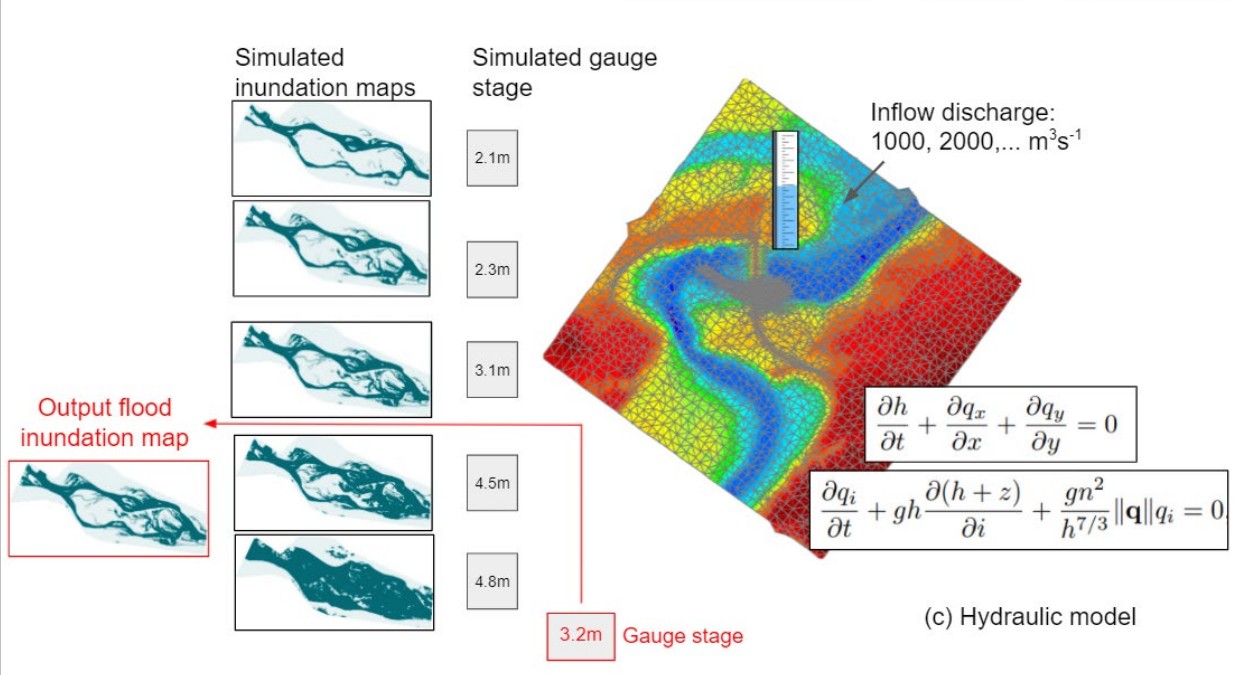

**Figure 3.** Schemes of the Thresholding (a), Manifold (b) and Hydraulic (c) flood inundation models



## 2.4. Alerts

Alerts are distributed to government authorities, emergency response agencies, and the affected population. The alerts include information about the forecasted water stage at the target gauge, the inundation map, and inundation depth if available. Flood
information is disseminated to the public (affected population) over three channels (Figure 4): 1) Google Search (for example, when searching for a flood at a specific location, or "flood near me" when located near a flood region); 2) smartphones within the forecasted flood inundated area receive a push notification on their phone when a forecast is issued; 3) icons presented in Google maps when viewing an area with an active flood alert. In all cases, the alert leads to a page with additional warning information, including the forecasted stage change at the target gauge and a map view of the inundated area and their depth, if
available. In addition, relevant organizations (government agencies and NGOs) receive flood alerts via email and specialized dashboards for their use; these likewise contain the forecasted stage and inundation map. Special care is taken to provide the alerts in the local language and to make it informative and actionable.

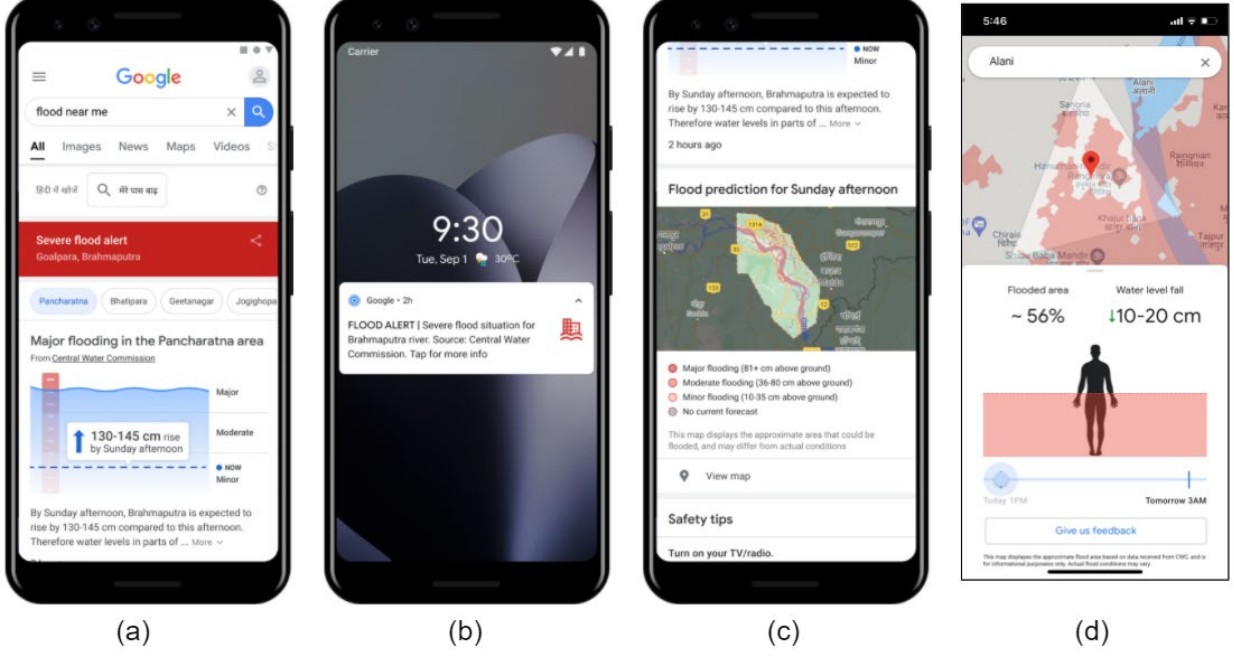

**Figure 4.** Examples of flood alerts sent to the public via different channels: (a) web-search; (b) smartphone push notification
(alert is shown at the local language); (c) Google Maps (© Google Maps 2021); (d) The Flood Hub, which is a user interface that contains the Google flood alerts per a single village or smaller area level (map data from © Google Maps 2021). It includes a zoomed-in map of the village, water depth and the percentage of the village that is covered by water. The river water level forecast is displayed as well.



## 2.5. Deployment in India and Bangladesh for the 2021 monsoon season

India and Bangladesh host the large river systems of the Indus, the Ganga, and the Brahmaputra, that provide livelihoods for hundreds of millions of people; but at the same time, they frequently flow over their banks, leading to loss of lives and substantial damage during the summer monsoon season (World Bank, 2015). In fact, it was shown that Asian river floods are the most devastating in terms of number of people killed and affected (Jonkman, 2005).

The Indian Central Water Commission (CWC), and Bangladesh Water Development Board (BWDB) provide flood forecasts and warnings in the two countries. The CWC bases operational forecasts primarily on a mathematical model that utilizes water stage data. Flood forecasts by BWDB are based on predicted river discharges at upstream boundary locations and a hydraulic model (World Bank, 2015).

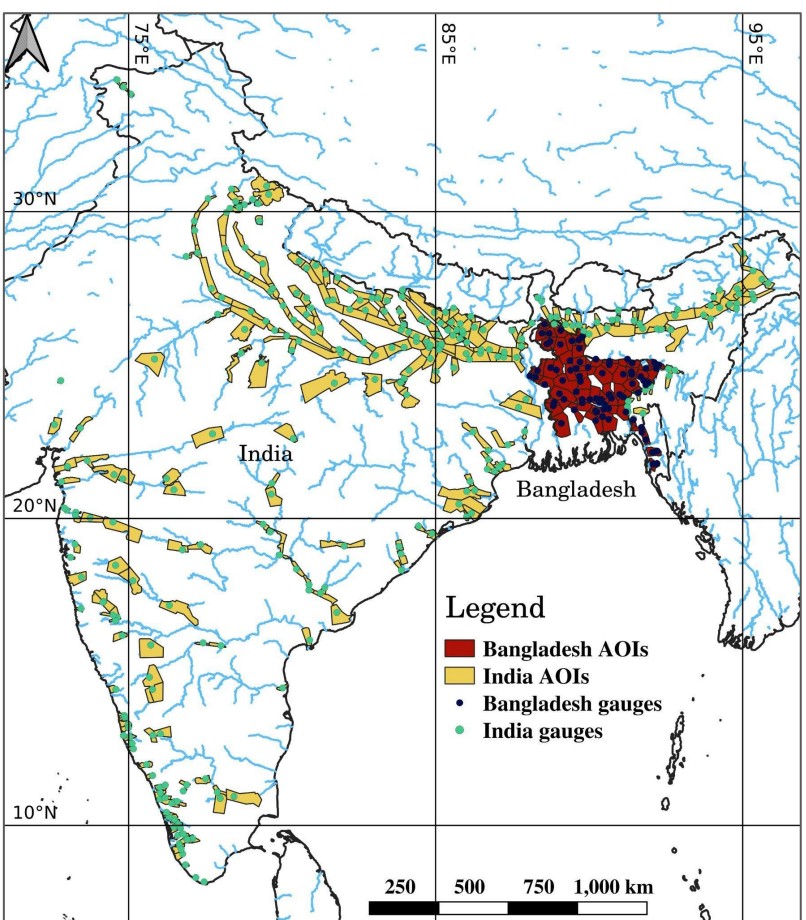

**Figure 5.** Target gauges (circles) and AOIs (polygons) for which the system was deployed in 2021 in India and Bangladesh.

Google's river flood warning system has been operational in India since 2018 and Bangladesh since 2020. These systems were expanded and modified for the 2021 monsoon season. During 2021 the flood warning system handled 376 target gauges,





covering watershed sizes of 350 to 1,500,000 km² (Table S1). Of these, stage forecast models were applied to 167 target

gauges, while the others were based on external forecasts. Inundation models were applied to 233 target gauges. Additional
143 target gauges, where a well-performed inundation model could not be applied, typically due to very limited ground truth
data or highly irregular flood pattern, were managed by the system but alerts were sent without inundation maps. In total
Google's flood warning system covered a flood-prone area of more than 287,000 km² with a population close to 245,000,000
(Table S1). Figure 5 and table S1 provide details on the target gauges and AOIs geographical distributions, stage forecast and

inundation model type, *maximal lead time*, watershed area, AOI area and estimated population size.

Stream gauge stage data for the monsoon season, i.e., June 1 to October 31, were obtained at hourly resolution from CWC and
BWDB. IMERG (Early Run) precipitation data were obtained from NASA and averaged in space and time to produce
watershed-averaged hourly data. Historical records for 2014-2020 were used to train the stage forecast model, which was then
used operationally in 2021 to provide forecasts using real-time data. As the LSTM models resulted in improved forecast skills

compared to the Linear models (see section 3.1 below) they were used for stage forecasting for all target gauges except two,
where LSTM performances were low. The specific configurations of the operational models are presented in Table 1.

**Table 1.** Stage forecast operational model configuration during 2021 monsoon season in India and Bangladesh

| Stage forecast model | Number of target gauges | Input | Output | *Maximal lead-time* | Comments |
|---|---|---|---|---|---|
| Linear | 2 | - Hourly water stages for the past 72 hours for the target and upstream gauges (typically 2-5 gauges) | Forecast stage at the target gauge for a given hourly lead time between the forecast time and forecast time + *maximal lead time* | 18 hours (Table S1) | Model is optimized for each hourly lead time up to the *maximal lead time* and for each target gauge separately |
| LSTM | 165 | - Hourly water stages for the past 168 hours for the target gauge and the past 240 hours for upstream gauges (typically 2-5 gauges)<br>- Hourly mean areal precipitation for the past 168 hours | Forecast stage at the target gauge for all hourly lead times between the forecast time and forecast time + *maximal lead time* | 8-48 hours (Table S1) | - There are 128 cell states in the hindcast and forecast LSTM<br>- Model is optimized for all hourly lead times up to the maximal, which is 48 hours for all gauges, and for all target gauges together |

The stage forecast models provide in real-time, at each hour, forecasts of hourly water stages between the current timestamp
and a future time, which is the gauge's *maximal lead time*. In addition, external forecasts for all 376 target gauges from the
operational models of CWC and BWDB were fed to the system each time they were available (typically, once or twice per
day with lead time between 12 and 120 hours). The forecasted stages were compared with the warning thresholds to determine
whether an alert should be issued. If so, the forecasts were propagated to the inundation component to generate inundation

maps.



The Thresholding and Manifold inundation models were applied to 174 and 59 target gauges, respectively, covering AOIs ranging from 20 to 5,295 km$^2$ (Figure 5, Table S1). The two models computed inundation maps (with the Manifold model also computing inundation depth) at a 16x16 m$^2$ spatial resolution, which were evaluated at a 64x64 m$^2$ spatial resolution. The models were trained based on historical flood events for 2016-2020 (a mean of 18 and a median of 4 events per AOI) and

utilized in real-time for 2021.

Alerts containing the forecasted change in water stage at the target gauge, the expected inundation map and water depth maps, if available, were sent to several agencies, including CWC, BWDB, and the Federation of the Red Cross and Red Crescent Societies. In cases where flooding was sufficiently severe, alerts were also sent as push notifications to smartphones situated in locations that were forecasted to be flooded. In total, about 115M notifications were sent and have reached about 22M

people (Figure 6). The notifications were sent in the local language, where nine different languages were used for India and Bangladesh, specifically English, Hindi, Bengali, Gujarati, Marathi, Tamil, Telugu, Kannada, and Malayalam.

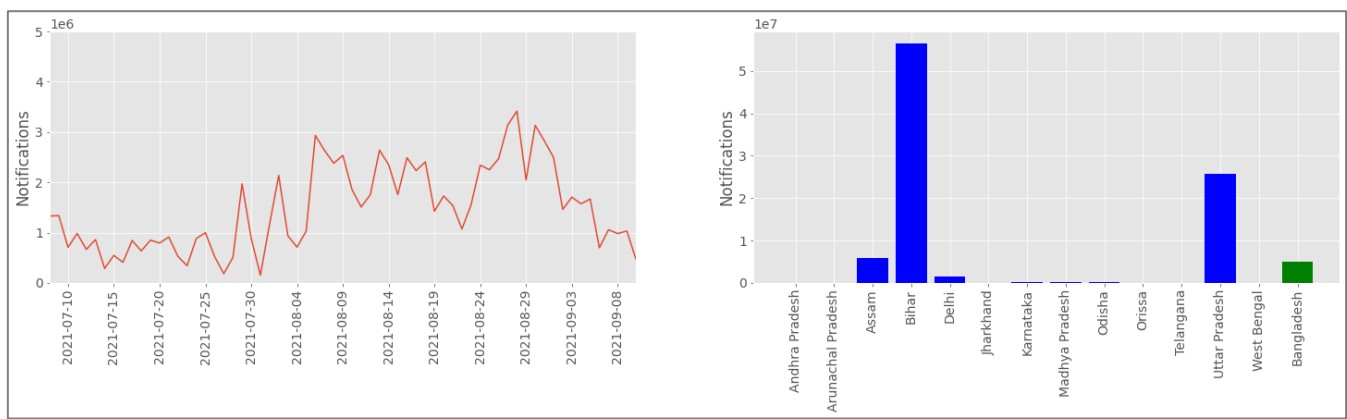

**Figure 6.** Total notifications sent from July 8 to September 10, 2021 along the monsoon season (left) and their distributions among states in India and Bangladesh (right).


As a demonstration of one flooding case we present in Figure 7 the system outputs for the flood event in mid-August in Bihar. On August 11, close to midnight, a few gauges in Bihar presented danger level alerts (Figure 7a). Specifically, the Hathidah target gauge had a forecast indicating that on August 12, 22:00 the gauge level would reach the highest recorded stage of this gauge (Figure 7b). The inundation water depth map (Figure 7c) covers an area of 533 km$^2$ (compared to 163 km$^2$ at low river

stages during the non-monsoon season). Also shown are photos from the affected areas (Figure 7d,e) and an example to the alert as published by CWC for this event (Figure 7f).



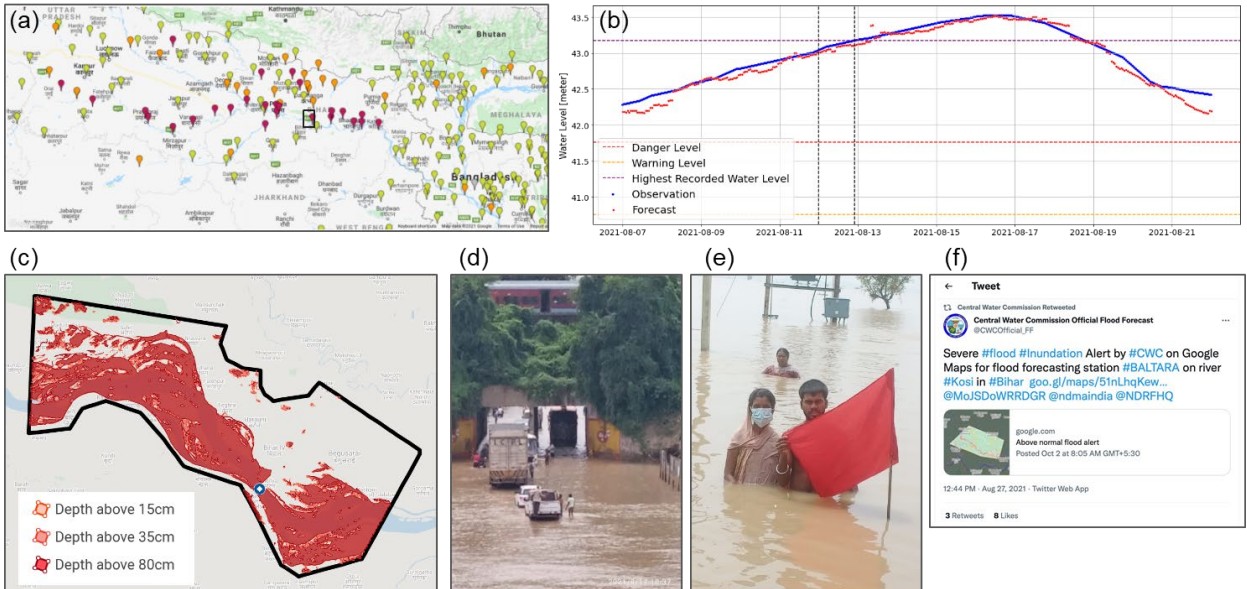

**Figure 7.** (a) Gauge alert map captured on August 11, 23:45, forecast for August 12, 22:00. Gauge colors represent: below warning level (green), above warning level (orange), above danger level (red) (map data from © Google Maps 2021). (b) 42-hour forecast (LSTM model) and observed river water stage for the Hathidah target gauge around the flood event. (c) The inundation depth map (Manifold model) forecasted for the Hathidah gauge at August 11, 23:45 referring to August 12, 22:00 (these two times are marked as vertical dashed black lines in b). (d) A photo taken during the flood on the way from Begusara to Patna (given to the authors by the Yunganter organization, www.yuganter.org.in). (e) A photo taken during the flood, Hetanpur, Madhopur Panchayat in the Patna district (given to the authors by the Yunganter organization, www.yuganter.org.in). (f) Alert sent for the flood event and published by CWC (Central Water Commission Official Flood Forecast, 2021).

## 3. Machine learning model evaluation

### 3.1. Stage forecast models evaluation

The stage forecast models presented in Section 2.2 are analyzed to assess their performances and to gain insights about the importance of their inputs. Accordingly, four models are examined:

- Linear model with past water stage input
- Linear model with past water stage and past precipitation input
- LSTM model with past water stage input
- LSTM model with past water stage and past precipitation input

The Nash-Sutcliffe efficiency (NSE) and the persistent-NSE metrics are used to represent the level of fit, defined as:





$1 - \frac{\sum_{i=1}^{n}(y_i^o - y_i^c)^2}{\sum_{i=1}^{n}(y_i^o - \overline{y^o})^2}$ and $1 - \frac{\sum_{i=1}^{n}(y_i^o - y_i^c)^2}{\sum_{i=1}^{n}(y_i^o - y_i^p)^2}$ , respectively, where $n$ is the series length, $y_i^o$ is the i[th] observed stage data, $y_i^c$ is the i[th]

computed stage data, and, $y_i^p$ is the last observed stage data at the time of forecast. NSE compares the model's sum of squared

error to the sum of squared error when the mean observation is used as the forecast value. Persistent-NSE, on the other hand,

400   uses the last observed data as the forecast value being compared against.

A 1-year cross-validation scheme was applied, based on 2014-2020 data. Figure 8 presents the distributions of NSE and

Persistent-NSE for the 167 target gauges with an operational stage forecast model (Table S1) and Table 2 provides their

statistics.

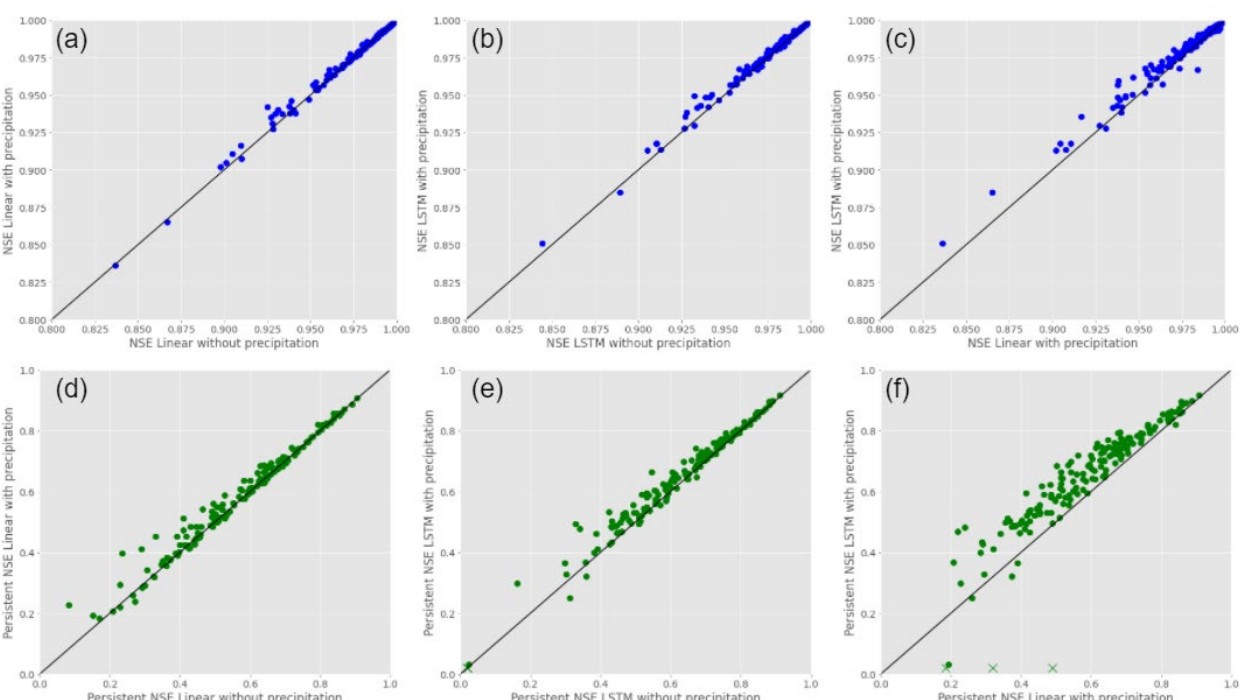

**Figure 8.** Comparison of the NSE (a-c) and Persistent-NSE (d-f) between: the Linear models without and with precipitation (a, d); LSTM without and with precipitation (b, e); and, Linear and LSTM with precipitation (c, f), based on data from 2014-2020, for 167 target gauges in India and Bangladesh (Table S1). For a better visualization, x-axis is limited to 0.8-1 for NSE and to 0-1 for Persistent-NSE (three gauges with Persistent-NSE of the LSTM models between -0.9 and -0.3 are shown with
410   X symbols). All analyzes were done with a 1-year cross-validation scheme, based on 2014-2020 data, where the mean among the years is presented.



**Table 2.** Median and range of metric values for the stage forecast and inundation models and median and range of metric differences for compared models

| Stage forecast model | NSE | | Persistent-NSE | |
|---|---|---|---|---|
| | Median | Range | Median | Range |
| Linear without precipitation | 0.9839 | 0.8368-0.9988 | 0.5851 | 0.0830-0.9050 |
| Linear with precipitation | 0.9847 | 0.8368-0.9989 | 0.5991 | 0.1842-0.9076 |
| LSTM without precipitation | 0.9862 | 0.8444-0.9992 | 0.6608 | -0.8824-0.9124 |
| LSTM with precipitation | 0.9870 | 0.8508-0.9992 | 0.6708 | -0.6079-0.9171 |
| Linear with - without precipitation* | 0.0001 | -0.0033-0.0173 | 0.0033 | -0.0328-0.1611 |
| LSTM with - without precipitation* | 0.0005 | -0.0048-0.0170 | 0.0153 | -0.0710-0.5403 |
| LSTM - Linear with precipitation* | 0.0017 | -0.0175-0.0216 | 0.0635 | -1.0971-0.2504 |
| | | | | |
| Inundation model | F1 1-year cross validation | | F1 leave-extreme-out | |
| | Median | Range | Median | Range |
| Thresholding | 69.01 | 41.67-92.49 | 76.74 | 2.48-96.68 |
| Manifold | 69.12 | 36.91-91.36 | 76.27 | 10.36-95.46 |
| Manifold - Thresholding* | -1.31 | -6.09-13.66 | -1.79 | -40.42-21.87 |

\* Statistically significant (p-val<0.05) according to the Wilcoxon signed-rank test for the paired samples

NSE metric values are very high for all target gauges and models, indicating in general very good predictive skill. It should be noted that these high skills are not surprising given that we are modeling large rivers with a typically slow response and that we use historic water stages as well as upstream water stages as model inputs. The Persistent-NSE metric values are lower than NSE, implying that forecasting with the last observed value achieves a significant improvement over forecasting with the mean observation, and, consequently, results in a baseline that yields lower scores of our models.

The metrics show an advantage of the LSTM model over the Linear model and of including precipitation as a model input (Table 2), and according to the Wilcoxon signed-rank test for the paired samples, all these comparisons are statistically significant. For 95% of the target gauges, the LSTM model achieves a better Persistent-NSE score than the Linear model. Including precipitation data as input improved both the LSTM (89% of the gauges) and the Linear (65% of the gauges) models. Metric values of all models are positively correlated with watershed area, as can be seen in Figure 9, presenting the median Persistent-NSE for three bins of drainage area sizes. The variance within the bin of the smallest watersheds (<10,000 km$^2$) is the largest and the medians in this bin do not significantly differ from the medians of the middle-size bin (p-val>0.1, based on Mood's median test); the two other median comparisons (largest watershed bin relative to the middle-size bin and the smallest size bin) are significant (p-val<<0.01).



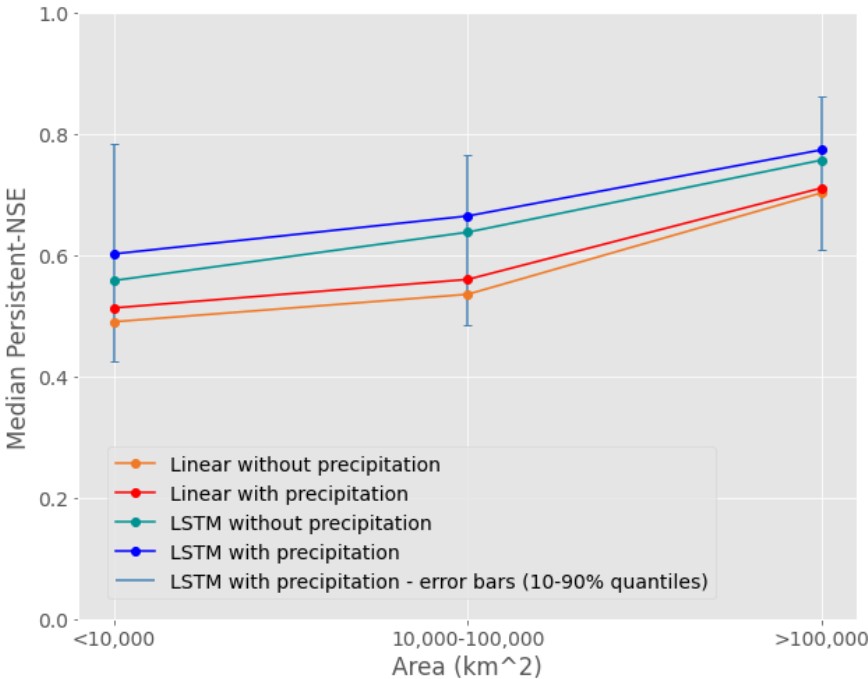

**Figure 9.** Persistent-NSE medians for three bins of watershed drainage area sizes. Number of gauges in the bins are 49, 54 and 42. Error bars are shown as the range of 10-90% quantiles in each bin. For a better visuality, they are only shown for the
LSTM with precipitation model. For all four models the median of the largest bin is significantly different than the median of the middle bin and the smallest bin (p-val<0.001, based on Mood's median test), while the difference between medians of the two smaller bins is insignificant (p-val>0.1).

### 3.2. Inundation models evaluation

The evaluation of the ML inundation models is done in two steps. First, we present a comparison of the Hydraulic model and

the Thresholding model. This step was important for benchmarking the ML models against more classical solutions. The F1

metric, which is the harmonic mean of the precision and recall, is used for evaluation, where the dry/wet classification of each

pixel in the AOI is examined and the wet pixels are taken as positive. Since the Hydraulic model requires substantial manual

work and heavy computation, the comparison was limited to 11 AOIs.

The two models (Section 2.3) were trained with flood events from 2016-2019 and validated with different events from 2020.

Table 3 shows the validation F1 scores achieved on the AOIs for each model, with the best score for each AOI in bold. The

Thresholding model outperforms the Hydraulic model on 9 of the 11 AOIs. In addition, the Thresholding model requires

significantly less computation and manual calibration effort than the Hydraulic model. Therefore, it was decided not to include

the Hydraulic model in the operational system.




**Table 3.** F1-scores for selected AOIs

| AOI | Hydraulic model | Thresholding model |
|---|---|---|
| Guwahati | 83.60% | **85.20%** |
| Gandhighat | 72.00% | **78.90%** |
| Goalpara | 72.20% | **81.20%** |
| Ayodhya | 73.00% | **75.10%** |
| Tezpur | 82.70% | **87.00%** |
| Neamatighat | 68.40% | **76.90%** |
| Kahalgaon | 65.00% | **76.70%** |
| Basua | 52.90% | **56.80%** |
| Turtipar | **70.60%** | 67.90% |
| Gangpur Siswan | **68.90%** | 66.60% |
| Balia | 65.90% | **73.60%** |
| **Average** | 70.47% | **75.08%** |

In the second step, we evaluate the Thresholding and Manifold models for all 126 AOIs with DEM (i.e., both models can be
applied), based on flood events from 2016-2020; in these years there are a total of 4815 flood events across all the AOIs (38
flood events on average per AOI). The median F1 (across years) is computed for each AOI, using a 1-year cross-validation
scheme (Figure 10a). In addition, we applied a "leave-extreme-out" validation procedure to estimate the model's skill at
accurately computing the inundation map for large unprecedented flood events that exceed water levels in the training sample.
In this procedure the training data set includes all events except the one with the highest recorded stage as well as all flood
events whose stage differs from the highest recorded stage by less than 30 cm. The trained model is then validated on the
highest flood event (Figure 10b). Table 2 provides the statistics of the F1 metric for the two inundation models and their
comparison.

The F1 metric has a narrower range for the cross-validation scheme, compared to the leave-extreme-out (Table 2). The
Thresholding model achieves better metric values than the Manifold model in a majority of AOIs (70% and 57% for the 1-
year cross-validation and leave-extreme-out analyses, respectively) and the median of the F1 metric differences (Thresholding-
Manifold), for both cases, are statistically significant according to the Wilcoxon signed-rank test for the paired samples (Table
2).





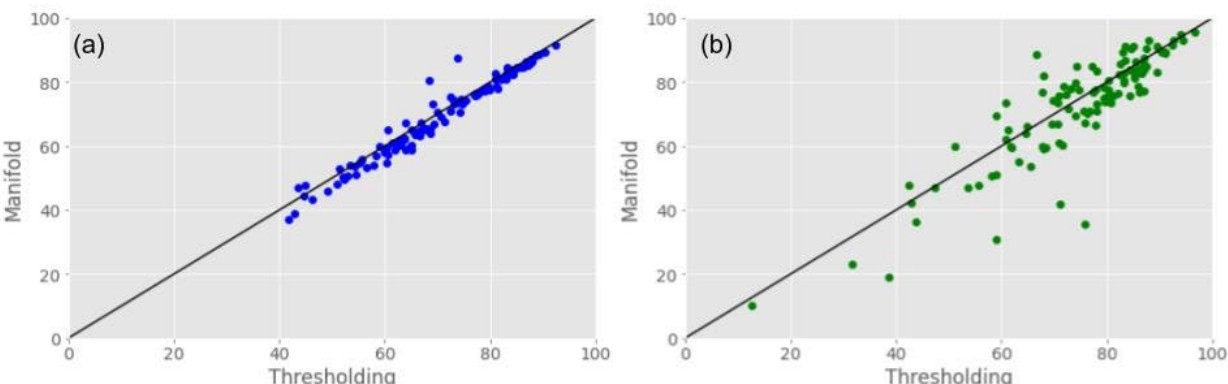

**Figure 10.** Comparison of the F1 metric between the Thresholding and Manifold inundation models for (a) 1-year cross-validation and (b) leave-extreme-out, based on data from 2016-2020, for 126 AOIs India and Bangladesh.

## 4. Discussion

This paper presents Google's operational flood warning system, already deployed in India and Bangladesh and planned to be further extended to other countries across the globe. It also presents ML models that are used in this operational framework for stage forecasting and for inundation mapping.

The benefit provided by the flood warning system to the region during the 2021 monsoon season was the relatively high temporal (hourly) and spatial (16-meter) resolution of the flood alerts for lead times ranging over 8-48 hours and over a large spatial extent (287,000 km$^2$ AOI area, Figure 5, Table S1). Approximately 115M flood alerts were sent to affected populations (Figure 6) and reached about 22M people. The accuracy metrics for both river stage and flood inundation area indicate good performance metrics (NSE and Persistent-NSE medians for the LSTM with precipitation stage forecast model are 0.99 and 0.69, respectively; F1 for cross-validation and extreme-leave-out are 69% and 76%, respectively; Figures 8 and 10).

The ability to compare performance to other benchmarks is limited, as we can only do so with the few published operational framework metrics. One of these studies by Zalenski et al. (2017) evaluates the US National Weather Service stage forecast models for 51 stream gauges in Iowa, US. They report root mean square error of water stage with medians of 0.3 and 0.5 meter (for below and above flood stage classes, respectively) for a lead time of 24 hours; increasing lead time to 48 hours increased these medians by 0.1 meters. The median of the root mean square error of the stage forecast models used in the present system are 0.1 and 0.2 meter for lead times of 24 and 48 hours, respectively. Although the errors are not fully comparable since these watersheds differ from those analyzed in Zalenski et al. (2017), they provide a good indication on their level of accuracy. The current paper presents accuracy metrics for operational stage forecast and inundation models for 167 and 126, respectively, stream gauges in India and Bangladesh and thus contributes to reducing the lack of information on operational models' performance and can be further used for evaluating other operational systems.





One of the challenges in operational flood forecasts is turning forecasts into effective warnings. Pagano et al. (2014) discussed
this challenge and presented the following concerns: "Is the intended audience receiving the forecasts? Is the information being
understood? Is the information being used to make the right decisions?". Regarding the second point, experience gained from
Google's system demonstrates the importance of using the local language for the alerts, in accordance with Perera et al. (2020)
who suggest community-specific warning messages composed by their recipients. Perera et al. (2020) also emphasize the
importance of targeted warnings, as opposed to warnings that are communicated to the entire population, which is the approach
taken here. Substantial efforts are put into finding effective ways to communicate flood alerts; for example, by providing alerts
at the local languages, accompanying the alerts with visual explanations, and investigating what text the alerts should include
to make them easier to understand. Two more directions are being explored for the presented system with respect to alert
efficacy. The first will allow individuals to define specific locations they are interested in (such as their village) and receive
more detailed information about those locations. The second concerns forecasting river stages at longer lead times, even at the
cost of increased uncertainty, which can be used to drive longer-term decisions; however, communicating such information is
an ongoing challenge.

ML methods are an integral and critical part of the river stage forecasting and flood inundation models. For stage forecasts, it
has already been shown that LSTMs outperform standard hydrological models not just for gauged basins but also for ungauged
basins (Kratzert et al., 2019a, 2019b), and even for extreme out-of-training events (Frames et al., 2021). For inundation
modeling, a comparison of the two ML-based inundation models with the physics-based hydraulic model found the ML models
exhibited higher accuracy (Table 1). Another advantage of ML models is that they require less manual per-gauge work, thus
enabling expansion to a larger coverage area. Indeed, it is generally easier to apply ML models to a new basin compared to a
process-based model: for example, Nevo et al., 2020, compared the hydraulic model and an ML inundation model and showed
that on average, the hydraulic model requires about 8 times more manual work; this is a result of the need for gauge-specific
parameter calibration in almost all process-based models (see Nearing et al., 2020 for some discussion).
On the other hand, there are potential drawbacks in ML models that need to be considered. These models rely solely on the
data and thus require large, representative records to reach a sufficient performance. Though one might expect that process-
based models would be more resilient to data limitations compared to ML models, due to the explicit process representation
and the physical meaning of some of their parameters (e.g., saturated hydraulic conductivity), in practice, there is no indication
for such advantage. In fact, many of the process-based parameters are scale-dependent (Beven, 1995); their use in the model
is not straightforward, and generally needs to be calibrated for reliable model performance. Furthermore, Moshe et al. (2020)
presents an ML hydrological model that can be trained with short records. In the present study we show that good performance
of the ML models is achieved for the analyzed basins with six to seven years of training data, a historical record length that
can be expected to be available in many other locations.





Google's flood warning system in its present form is designed for large gauged rivers. In the future, we plan on extending the system to accomplish two additional goals:

Ungauged basins: Prediction in ungauged basins is one of the main challenges in hydrological sciences (Sivapalan et al., 2003),
and despite significant research activity and advances in this direction, robust and reliable flood predictions in ungauged basins are still lacking (Hrachowitz et al., 2013). The flood warning system presented here focuses on gauged basins and relies on past measurements from the target stream gauges, both for training and for real-time model input. Extending the system to ungauged locations implies that precipitation becomes the most dominant input. Moreover, in this case the models cannot be optimized to the target basins. Although these are major obstacles, encouraging results from recent studies with a similar
LSTM model (Kratzert et al., 2019a) have shown that relatively good prediction can be achieved in ungauged basins when the model is trained on gauges from other locations in the region.

Flash floods and small-to-medium-sized basins: The system described in this paper was developed for floods in large rivers. We aim to expand the technique to small-to-medium basins ($< 1{,}000$ km$^2$) and to handle flash floods, which appear only a few hours following an intense storm (Borga et al., 2011). This extension implies a much faster response, which in turn implies
less information in the past stage data and more weight to be put on forcing input data, most importantly precipitation. Furthermore, uncertainties in precipitation data might significantly affect the flood forecasts and must be taken into account (Borga et al., 2011).

## 5. Conclusions

Operational flood warning systems can save lives and reduce risks and damages. Google's system was developed to provide accurate and effective warnings for floods in large gauged rivers where a dense population is located. After operational deployment in India and Bangladesh, the following conclusions can be drawn:

- During monsoon 2021 the system provided over 100 million flood alerts directly to individuals who were located in the flooded areas, as well as to relevant agencies.
- Two ML models are used for river water stage forecast: Linear and LSTM. The LSTM model was significantly better than the Linear model (0.67 vs. 0.60 medians of Persistent-NSE) and therefore was the one used operationally in 2021 for 165 out of 167 target gauges.
- The input to the stage forecast models was the past observed water stage at the target gauge (168 hours for the LSTM model) and a few upstream gauges (240 hours for the LSTM model). Adding past precipitation data (hourly mean
areal precipitation for the past 168 hours from IMERG early run) slightly improved model performance and therefore was used operationally.
- Two ML models (Thresholding and Manifold) were developed for inundation mapping. Furthermore, the Manifold model, using DEM data, additionally provides forecasts of water depth in inundation areas, which is considered



crucial information for the affected individuals. The Thresholding model was found to outperform the physics-based hydraulic model and requires significantly less manual effort and computational resources which allow its application on a large scale. The Thresholding model also has a higher skill compared to the Manifold model although the distribution of their F1 metric is close (F1 medians of 69 and 76% in leave-1-year cross-validation leave-extreme-out schemes). Both models were applied operationally during 2021.

- Continually improving the effectiveness of flood alerts is an important and ongoing challenge. Feedback from people in the flooded regions has helped increase alert effectiveness. For example, we found that it was important to provide the alerts in the local language, and to notify of the expected change in water stage (and whether it is expected to rise or fall) rather than the absolute stage.

- The literature lacks performance results from operational flood warning systems. Such results could contribute to developing common scientific knowledge and improving flood warning systems. We hope that the results presented in this study will be helpful for future studies assessing operational systems.

**Code availability**

Software was developed at Google and the code is proprietary.

**Data availability**

Hourly stage data were received from the Central Water Commission (CWC), India, and the Bangladesh Water Development Board (BWDB). These data are government property and cannot be distributed. Precipitation data were obtained from the National Aeronautics and Space Administration (NASA) at https://gpm.nasa.gov/data/imerg (product: GPM IMERG Early Precipitation L3 Half Hourly 0.1 degree x 0.1 degree V06, GPM_3IMERGHHL 06). Historical flood extent data was derived from Sentinel-1 SAR GRD data, which is publicly available on Google Earth Engine (Image Collection 580 COPERNICUS/S1_GRD).

**Author contribution**

The flood warning system presented in this paper was designed, developed and programmed by all co-authors. Paper writing was led by EM and reviewed and edited by all co-authors.

**Competing interests**

Some authors are members of the editorial board of the journal Hydrology and Earth System Sciences. The peer-review process was guided by an independent editor, and the authors have also no other competing interests to declare.

**Acknowledgments**

Many people participated in researching, developing, and deploying Google's flood forecasting system. We would like to particularly thank Aaron Yonas, Abhishek Modi, Aditi Bansal, Adrien Amar, Ajai Tirumali, Anshul Soni, Asmita Metrewar, 590 Bernd Steinert, Bradley Goldstein, Brett Allen, Gopalan Sivathanu, Jason Clark, Joanne Syben, Karan Agarwal, Kartik Murthy, Kay Zhu, Lei He, Manan Singhi, Mark Duchaineau, Matt Manolides, Mor Schlesinger, Novita Mayasari, Paul Merrell, Rhett Stucki, Ruha Devanesan, Sandeep Kotresh, Saurabh Rathi, Sergey Shevchenko, Slava Salasin, Stacie Chan, Subramaniam Thirunavukkarasu, Tal Cohen, Tom Small, Tomer Shefet, and Zhouliang Kang for their contributions.
We thank the Central Water Commission (CWC), India, and the Bangladesh Water Development Board (BWDB) for providing 595 data and for their collaboration.



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
