# Peer review of "Flood forecasting with machine learning models in an operational framework"

_Hydrology and Earth System Sciences, 2021_

## Referee Comment (RC3)

[referee-annotated manuscript omitted]

---

## Author Response (AR1)

**Response document**

We are thankful for the constructive comments we received from the three reviewers. We addressed all comments and prepared a revised paper which we believe is much improved. We thank the editor in advance for further handling the paper. Below is a detailed response for each of the review reports.

**Response to reviewer #1**

We thank the reviewer for the time and efforts in reviewing the manuscript. Below is our response where the reviewer comments are in black, our response is in green, and the copied text from the manuscript is in blue.

This paper presents a complete workflow for an operational flood forecasting and mapping case study.

The paper details all the critical steps in an operational system for data scarce regions, including remote sensing data integration, forecasting, inundation mapping, and communicating forecasts to the public.

We thank the reviewer for this encouraging statement.

The model performance comparison against linear regression and simple hydraulic models is not a very challenging task compared to evaluating performance against better performing ML and physical models cited in the paper.

As a start, it is important to emphasize that almost all physically-based, process-based, and even most of the ML-based, hydrological models in research and in operation are not using past water stages as their input, while for our models this is the main input data. This input makes a major difference for large gauged rivers (which is the focus of our system) and increases the prediction accuracy significantly. Even the Linear model, with past river stage input, has a median NSE >0.98. Therefore, it is not a "fair" comparison to evaluate our stage forecasting models against other hydrological models that do not use the past stage data as input. We realize this is an important point that needs to be better emphasized in the paper. Therefore, we added the following paragraph to the revised paper at the beginning of Section 2.2 (Lines 131-134):

"The primary input to the stage forecast models is past river stages (alongside complementary information such as weather forcing) and the output is the future river stage. It should be noted that in this aspect they differ from many standard hydrological models where the input does not include river stages but mainly forcing data and watershed properties."

As an ML stage forecast model we adapted the LSTM. This model is the most accurate one in the published literature to our operational use case, with some modifications that were necessary. The Linear model, which was used in earlier operational versions, was kept for very few locations where the LSTM results were not sufficient. As already explained above, even the relatively simple Linear model shows high skills (Table 2, Figure 8) and, as we state in the paper, this is "...not surprising given that we are modeling large rivers with a typically slow response and that we use historic water stages as well as upstream water stages as model inputs". Therefore, it was important to show the added value for the transition from the Linear model into the, much more complex, LSTM model. For this reason these models are compared.

The physically-based Hydraulic model presented in the paper was used operationally in earlier versions of the warning system (it was operational in India during the 2018 and 2019 monsoon seasons). The model is based on the numerical scheme of the hydrodynamic model proposed in de Almeida et al. (2012) which is a modification of the simplified inertial formulation for the 2D shallow water model (Bates et al.,2010). Such hydraulic models constitute the vast majority of existing operational inundation models (e.g. using DHI's MIKE, Deltares' Delft-FEWS, etc.) and was a good choice for our operational framework that involves hundreds of stream gauges and river stage data (rather than discharge).

While used operationally, large efforts were made to use the Hydraulic model with the best parametrization and to improve its accuracy. But, because the comparisons described in this paper showed the new ML-based inundation models to be superior in both accuracy, scalability, and computational costs, the Hydraulic model is currently not operational. It was important to demonstrate in the paper the improvement in accuracy achieved with the ML inundation model, compared to the ubiquitously-used Hydraulic model, and this is the reason we included this part in the paper. True, there are other physically-based hydraulic models that can be tested against the ML inundation models we have used, and possibly their accuracy would be higher. But given the scalability limitation and computational cost of such models, we do not think adopting these models to our system and adding more comparisons with those models would lead to any important findings that justify such large efforts.

The performances of our models can be judged by the metrics we present for the validation data sets in terms of forecasted stages and flood inundations (Table 2, Figure 8, Figure 10). Both are showing good results. We were looking for other publications with metrics from

operational systems, in order to evaluate our system, but there are only a few studies that include such metrics. We refer to this issue in the discussion.

The spatial and temporal resolution of the input data for rainfall, elevation, and other datasets is rather low to achieve comparable forecasts, but still useful in data scarce regions with limited resources.

For rainfall in data-scarce regions, satellite-based products are the most practical option. We are using the IMERG data that has an original resolution of 0.1° and 30-min, which is among the highest available for real-time or near real-time global rainfall products. The rainfall data are further averaged in space over the watersheds area and in time to match the hourly resolution (for the implementation in India and Bangladesh). While this lowering of spatial resolution may be debatable, we emphasize two important points: 1) as many hydrological studies have shown, there is a tradeoff in spatially averaging precipitation data. Surely, floods are sensitive to the spatial patterns of precipitation but at the same time unsystematic errors in this (often inaccurate) information are reduced with averaging; 2) as opposed to standard hydrological models, where precipitation is the most important input, for the present system the main input is past river stages (from the target gauge and its upstream gauges). Precipitation input improves the prediction accuracy, on top of the accuracy achieved with past stages (Table 2, Figure 8), but the model is already quite accurate even without precipitation input. In this situation, it is better to keep the precipitation spatially averaged, since adding many precipitation features would have considerably increased the complexity of the model.

Elevation data, used in this system for inundation modeling, is actually of quite a high resolution compared to other publicly available global DEMs. We use a 1-meter resolution DEM, while SRTM is a 30-meter resolution DEM. Lidar-based DEMs can get to a much higher resolution but these are not currently available globally. The high-resolution DEM is produced by Google and is being kept up to date. Through training and validation of the inundation models, it was found out that 16 m resolution achieves similar accuracy to using higher-resolution versions of the DEM but at vastly lower computational costs.

Following the reviewer's comment, we noticed that we did not write the base resolution of the DEM. We added these details in the revised manuscript in Section 2.3 under DEMs for target gauge AOIs (Lines 315-316):

"Consequently, higher-resolution (1x1 meter2), up-to-date DEMs are constructed for each AOI…"

Some of the details, like how AOI pixels are defined in the flooding regions, are not clear in the paper.

Indeed this detail is missing. We added to the revised paper the following text about AOIs (as a part of Section 2.5, Lines 363-365):

"AOIs for the target gauges were defined as polygons around the river at the gauge location. They were computed, by both manual and automatic procedures, by observing the flood patterns in the area around the gauge from historical flood inundations maps."

We also added some missing details requested by reviewer #2 about training and validation data sets and number of flood events in training and validation, which we are highlighting here as well for context, as follows:

In Section 2.2 (Stage forecast modeling) under "Training and validation" we added (Lines 185-187): "The training and validation data sets are composed of samples where the features are past river stages at the target and upstream gauges and past spatially-averaged precipitation (for the LSTM model). The labels for training are future river stages at the target gauge for a given lead time."

In Section 2.3 (Inundation modeling) under "Training and validation" we modified the present text to clarify the structure of the training and validation data sets (Lines 297-299): "The training and validation data sets for the inundation models are composed of samples representing historical flood events where the features are gauge water stage measurements and the labels are the corresponding flood inundation extent maps from satellite data."

We provided the numbers of flood events used for training the ML inundation models for the 2021 implementation in Section 2.5 (although they were corrected in the revised manuscript). But we did not provide the numbers for the cross-validation analysis in Section 3. We added these details in the revised manuscript (Line 487-488): "...in these years there are a total of 4815 flood events across all the AOIs (on average 34 events were used for training and 10 for validation per AOI)."

**Response to reviewer #2**

We thank the reviewer for the time and effort and for the detailed review report that helped us improve the paper. Below is our response where the reviewer comments are in black, the response is in green, and the copied text from the manuscript is in blue.

This paper presents ML models that respectively (i) directly predicts the river stage (rather than predicting discharge and then translating to stage); (ii) predict wet/dry of pixels depending on gauge stage; (iii) and estimate flood inundation depth. Among these, (i) was trained based on historical stream gauge data and near-real-time upstream gauges; (ii) was trained using historical satellite data and coincident stream gauge height data. (iii) was not really a model, per say, but an interpolation procedure.

I think the paper demonstrated strong performance from a completely data-driven model. It highlights the idea of directly simulating stream gauge height, which breaks many barriers. If they didn't do this, they need to simulation discharge and then resolve the highly-variable (in space) relationship between discharge and stage height. Most of time we cannot resolve it. In the authors' case, there is no discharge data to begin with. So directly tackling gaging height is a good and necessary idea (but it also leads to some issues I will discuss below). The paper also demonstrates a very efficient forecasting scheme based on upstream gauge data. The whole paper demonstrated how to stack different models together. The authors also showed a unique flood inundation component that is accurate. The work is very useful for hundreds of millions of people and it takes lots of courage to take on such a responsibility.

We thank the reviewer for highlighting this paper's contributions.

While there are many reasons why I like this paper and I encourage the publication of this paper, I also noticed a few major issues. These issues are raised here in the hope to make the manuscript more balanced and comprehensive.

Thank you for pointing out these issues. Your comments are very helpful and helped us to improve the paper. Please see below our point-by-point response to each of them.

(a1) There should be some discussion of the potential scientific limitations (even if caused by practical data availability) of the approach and the conditions under which this approach is applicable. As far as I can see, all the models were posed in a highly case-specific way. The gauge height LSTM model has weights that are shared across multiple gauges but it

also needs gauge-specific weights that are tuned to local data with a particular configuration. (how much worse will it get if you don't use those gage-specific weights?) The inundation extent model is tied to the gauge and the particular river bathymetry downstream from that gauge. In other words, it seems these models can only be applied where gauge data is available for training. The trained relationship is not portable anywhere else (if so, it poses a requirement on the available data records). Don't get me wrong. I think the model is highly useful operationally. In India there are many places where the model is applicable. It just might make sense, if these limitations are true, authors can discuss where and when this model formulation is valid so it is easier for the readers to understand if these algorithms are sound for their purpose. Maybe they can come up with a more uniform model and show its accuracy.

Thank you for this comment. Indeed, as we already emphasize in the paper, and better clarify in the revised version, this model is aimed at gauged locations in relatively large, slow-responding rivers. Indeed, the LSTM model training has both shared and gauge-specific weights. Gauge-specific weights are required for incorporating upstream gauge data as input, which is a very informative input. Without gauge-specific weights, the LSTM performs much worse. It is also true that the inundation model is also gauge-specific. This indeed requires having some data for training (past stage data and flood inundation from satellite). We do not claim these weights can be transferred to other locations, only that adding new locations which meet the above conditions is easy, or in other words, the models are scalable. See our response to the next comment concerning scalability.

We agree the limitations need to be stated clearly and therefore we added the following paragraph to the discussion section in the revised manuscript (Lines 571-577):

"The presented system has some limitations that need to be emphasized. First, Google's flood warning system in its present form is designed for flood forecasting in gauged river locations. Specifically, this requires, at a minimum, river gauges providing their stage data in real-time that have records of historical stage data and a few cases of flood events to be used for training and validation of inundation models (here, records of 6-7 years were used). Second, the stage forecasting and inundation models would work best in slow-responding, large rivers. Apparently, even with these restrictions, such flood warning systems can be useful for a large population worldwide. We are currently working on addressing these limitations by extending the system to accomplish two additional goals:..."

Actually, our plans (which are already underway) include expanding the model to ungauged locations, and improving performance on flashier rivers. This is already explained in the discussion section in the previous version of the paper.

We are not sure we understand what the reviewer means in a "more uniform model". If the meaning is a model with the same configuration for all gauges then our model indeed has the same configuration across all gauges, albeit with some parameters that are gauge

specific. This is similar to any hydrological model that has some watershed-specific parameters. If the reviewer meant something else, we would love to better understand and address the concern.

(a2) This point also contradicts the authors' claim that the model is highly scalable. You cannot take the model to a new terrain and directly apply it. In addition, the learned relationships may not always stand --- what if you have heavy rainfall in the region between your upstream gauge and the gauge of interest? It seems your model cannot consider such forcings (this may not matter that much for large-scale Indian monsoons, but it could be important elsewhere). This means, while the model is fast to run, it is not scalable in the sense of expanding to new areas ---- you must spend the time and effort to collect the data and train the model in every new area of interest, and that is assuming you are lucky enough to have the  data. Hence, it is uncertain how the authors intend to use the model on large areas.

Thanks for raising this point. Please note we did not use the term "highly scalable". The term "scalable" appears two times in the paper, one in the introduction describing the findings from previous studies and the second when we explain that the ML inundation models were found more scalable than the Hydraulic model. Nevertheless, we do claim the ML models described in this paper are generally scalable in the sense explained below.

We distinguish between scalability in the sense of applicability to different regimes and scalability in the sense of how easy it is to deploy the models at enormous scales within the applicable regimes. Here we refer to the latter, while for the former we clarified above (response to comment a1) what conditions our model is applicable to. When comparing scalability of our model to other hydrological and hydraulic models the tradeoff with accuracy needs to be remembered. Many hydrological and hydraulic models were developed to work with static (e.g., DEM, soils, landuse) and meteorological forcing data (e.g., precipitation, temperature) as input. If no calibration is needed, such models in principle have high scalability. However, without any calibration the accuracy of those models is very low, as was discussed in many hydrological publications. If high accuracy is needed, calibration against observed records is required, and this is known to be a demanding process. In particular operational frameworks require high accuracy models and thus models' calibration is an essential and important step in their deployment. We compare to those models when claiming an improvement to scalability.

As a measure for scalability, one can assess the time and effort it takes to deploy the system to a new large region (e.g., country, with hundreds of gauges), within the applicable regime. Requirements for data collection and quality control are identical for all models using gauge data and are therefore not included in the assessment. For the presented flood warning system we conservatively estimate that the ML stage forecast and inundation

models deployment would take for such a new region all together about 6-7 days of CPU time and manual work. On the other hand, the deployment and calibration of standard hydrological and hydraulic models to hundreds of new gauges would most probably take significantly longer.

It is also important to emphasize that the flood warning system is certainly intended to cover large areas. First, the system in its current version already covers 470K sq. kms and a population of more than 350 million, and is being extended to several new countries (in Asia, South America and elsewhere), including some that are not in the Indian Monsoon climatic regime. In parallel, we are actively developing a modified generalized global model which can be deployed both in gauged and in ungauged locations. This effort is mentioned in the paper, but we do not yet have results for this model to report.

Concerning the reviewer's point of heavy rainfall between the upstream gauge and the target gauge: although a situation as the reviewer describes can happen (though not commonly), we think the model is set up to be able to handle most of these cases as well. The main reason is that, in addition to the input from upstream gauges, the model input includes the watershed-averaged precipitation and the target gauge past stages. The situation the reviewer describes can typically happen when the upstream gauge is at a relatively large distance from the target gauge, which implies the additional watershed area between the upstream and the target gauges cannot be too small. For such a rainfall event to cause a large rise of the river stage at the target gauge location (especially in the large rivers we address) it must produce substantial rain amounts over this downstream watershed area. It is therefore probable this rainfall would produce some signal in the watershed-averaged precipitation, even though it does not cover the entire watershed. It is also probable that the target gauge will start rising due to the rainfall event. The signals in both inputs (i.e., precipitation and past target gauge stage) are likely to lead to a forecast of a general river rise. It is true such an event would have a shorter response time compared to more spread rainfall and it is possible such an event can be forecasted only with shorter lead times than the lead time selected for the gauge - yet it's worth noting that our system includes support for automatically shortening lead times in such events. The scenarios above are currently speculated and analyses are required to confirm their validity.

(a3) It also exerts some constrain on the eligibility of sites. Because you have to train a site-specific model, you can only use sites with long-enough records to train the model. The model cannot be large, and information from other sites do not help with a particular gauge of interest.

This is true but the required record length is not large. In this study, the training data set included data for 7 years for the stage forecast model and 5 years (i.e., flood events from this period) for the inundation model. The cross-validation analysis (presented in Section 3)

was based on training records of 6 and 4 years for the two models, respectively. We clarify this point in the new paragraph we added to the discussion Section (see text in the response to comment a1 above).

"The model cannot be large": this is true, ensuring the model does not overfit to the data is a key concern. The architecture presented is built specifically to address this concern - the shared weights among all locations allow for a much larger and more complex model (since it is trained across sites), while the per-gauge weights allow for some site-specific customization but are kept small. Our validation and test results show that the models do not overfit to the data.

"information from other sites do not help with a particular gauge of interest": we do not agree with this statement. The LSTMs are shared models, and as such allow the model for each target gauge to benefit from the training data of all gauges. This point relates to the previous ones - each specific gauge does not have enough historical training data to train a model as complex as LSTM - these models aggressively overfit when trained on individual gauges. However, the shared LSTM model allows the architecture to model complex rainfall-runoff patterns without overfitting.

(a4) If my understanding is incorrect, I stand corrected and the authors can show a test case where the model is applied to an "ungauged" location.

The reviewer is correct. This flood warning system indeed focuses on gauged locations. We wrote this in the previous version of the paper but better emphasize this in the revised version with the new discussion paragraph presented above.

(b) The training dataset for the models were not clearly described. For the inundation extent model, there should be descriptions of how many events were included as training and test images.

Thank you, these details were added to the revisions as follows:

In Section 2.2 (Stage forecast modeling) under "Training and validation" we add (Lines 185-187): "The training and validation data sets are composed of samples where the features are past river stages at the target and upstream gauges and past spatially-averaged precipitation (for the LSTM model). The labels for training are future river stages at the target gauge for a given lead time."

In Section 2.3 (Inundation modeling) under "Training and validation" we modified the present text to clarify the structure of the training and validation data sets (Lines 297-299):

"The training and validation data sets for the inundation models are composed of samples representing historical flood events where the features are gauge water stage measurements and the labels are the corresponding flood inundation extent maps from satellite data."

We provided the numbers of flood events used for training the ML inundation models for the 2021 implementation in Section 2.5 (although they were corrected in the revised paper). But we did not provide the numbers for the cross-validation analysis in Section 3. We added these details in the revised paper (Lines 487-488): "..in these years there are a total of 4815 flood events across all the AOIs (on average 34 events were used for training and 10 for validation per AOI)."

(c) It is not clear if the model accuracy drops as we go further downstream from the gauge. Some exploration here will be useful.

It is not clear if the reviewer refers here to the stage forecast model or to the inundation model. The stage forecast is only for the gauge location so we assume that the reviewer's comment refers to inundation modeling.

In response to the reviewer's comment we examined the relations between F1-score and the distance from the target gauge. Results, shown in the figure below, are taken from the training data set over the main Ganges and Brahmaputra AOIs that were launched in 2021. Every point in the graph is the average F1 over a ring of width 4 km from the gauge over all the events and AOIs (at least 200 cases per bin). Standard errors are also shown.

It should be noted that the F1 metric is biased toward regions that have larger fractions of permanent water (and thus more true positives). This bias favours bins that are closer to the gauge, and specifically the bin of 0-4 kms, which can partly explain its higher F1-score. It is worth saying that every metric has its own biases, so it is not clear whether there is a better metric to represent the fit.

Although there is some negative trend in the F1-score with distance, given the possible bias of the first bin and the variations of the metric in the rest of the bins, we cannot point to a clear drop in accuracy with distance from the gauge.

[Figure]

(d) regarding authors' criticism on the hydraulic model --- are we sure you feed it the best parameters and inputs? There is no description about calibration. Back to point (a), in a region without past observations, the hydraulic model may still function but the ML inundation model may not --- which means these models have their own use cases. If I'm wrong please correct.

Indeed many efforts were made to calibrate and benchmark the Hydraulic model, especially during the first years of the project when it was a part of the operational system (until 2019). We have tested our calibrated hydraulic models against standardized benchmarks, and paid well established third-party hydraulic engineering companies to set up and manually calibrate hydraulic models using standard platforms (e.g. Delft-FEWS) on our areas of interest - achieving similar metrics and forecasts to our own hydraulic models. Our analysis has also shown that the inundation pattern from the Hydraulic model is most sensitive to the upstream discharge and the downstream normal slope, while the effect of roughness coefficients on the inundation was much weaker and interacted with the other factors. Therefore, we ended up fixing the roughness coefficient per pixel (one of two values, representing either a river-bed or a non-river-bed pixel) and optimized the upstream discharge and, if needed, also the downstream slope. We ran about 1000 simulations per AOI to optimize those parameters. One cannot guarantee that those are the best parameters, as the optimization of such models is a complex process. But given all simulations, we assume the optimization is quite good and it is fair to compare the optimized Hydraulic model with the ML models. The reviewer is correct that we do not elaborate on the Hydraulic model optimization. We do so for the sake of conciseness since we are not using it in the operational system.

The reviewer raises a fair point that the Hydraulic model could be useful for locations without data on past flood inundations, while the ML models are not applicable in such cases. This would be correct in cases where the gauge input is discharge, rather than the

stage. However, with stage input data the Hydraulic model must be optimized for the relation between stage and inundation pattern and therefore requires records of historical flood inundation data and their respective gauge stage measurements, similarly to the ML models. As the reviewer points out in this same comment, the Hydraulic model often requires some level of calibration, unless some regional parametrization can be used. Ideally, if high quality, high resolution DEMs (including bathymetry) and real-time discharge data are available and if regional parametrization is applicable, Hydraulic models can indeed perform well. But in reality, this is a rare situation, and in most gauged sites the reliable application of the Hydraulic model, same as ML models, would require historical flood data.

(e) there seemed to be no description of network configurations such as hyperparameters, hidden size, minibatch (maybe there is not a minibatch), training epochs, etc.

Thank you. A new table (Table S2) was added in the Supplementary with hyperparameters and settings of the LSTM model.

(g) does it make sense to average precipitation for a drainage area > 100,000 km$_2$?

This is a good point. As many hydrological studies have shown, there is a tradeoff in spatially averaging precipitation data. Surely, floods are sensitive to the spatial patterns of precipitation but at the same time errors in this often inaccurate information are dumped out with averaging. However, as opposed to standard hydrological models where precipitation is the most important input, for the present system the main input signal is from past river stages (from the target gauge and its upstream gauges). Precipitation input improves the prediction accuracy, on top of the accuracy achieved with past stages, as shown in section 3.1, but the model is already quite accurate even without precipitation input. In this situation, it is better to keep the precipitation spatially averaged rather than adding many precipitation features and increasing the model complexity. Even when averaged over such large areas of hundreds of thousands square km, the mean-areal precipitation time series has value as it is correlated with the main precipitation systems that lead to an increase of these large river stages. It might be that in a very different climatic regime, where precipitation systems are substantially smaller (e.g., in the arid regions), or if past river stage data are not used as input, such an average would not make sense and another strategy would be taken. Indeed, in our efforts towards extending the models to more climatic regimes and to ungauged locations we are currently exploring averaging precipitation over sub-basins, but cannot report the results yet.

(h) We have no intuitive understanding of what F metrics mean. Do you mind showing some observed vs simulated maps for different values of the F metric?

Below are two maps for the same flood event with two different models. On the left, a trivial "model" simply assumes an inundated circular area, while the model on the right is a result of one of our inundation models. The colors represent true-positive (hits) in blue, false-positive (false-alarm) in yellow, false-negative (miss) in red, and true-negative in gray. The map on the left has: precision=53.8%, recall=16.4%, F1=25.1%. The map on the right has: precision=76.3%, recall=85.4%, F1=80.6%. Note that the map on the right has some gray area that was cut out, but true-negatives are not included in the computation of the F1-score so it does not affect this metric. We hope this is helpful. We added these examples to the supplementaries of the revised paper with a reference in the text (Figure S2, Supplementary).

[Figure]

(i) the flooding depth model was never tested and we do not know its accuracy. Can you talk about its value in the real world? Also, low-resolution could also give you discontinuity.

We agree with this point. In fact, we are putting considerable effort to collect ground truth depth data and get feedback from people in the affected regions via surveys in collaboration with local organizations. Unfortunately, the data we collected so far is too limited and too scattered - due to COVID limitations and other hurdles. Therefore we cannot yet present its accuracy, as we emphasize in the paper. We are continually investing in collecting more data to validate flood inundation and depth. To better clarify this point we added the following discussion paragraph to the revised paper (Lines 535-546):

"A few studies examined the effectiveness of flood alerts in operational frameworks. For example, Rotach et al. (2009), as a part of an end-to-end operational flood warning system in the Alpine region, collected feedback from end-users through questionnaires, interviews,

and workshops, and some initial insights are given on the utility of the system for the decision-makers and how well the information was perceived. It should be noted, however, that the current knowledge about effective flood warnings in countries like India and Bangladesh is very limited. For example, a literature review by Keller et al. (2021) shows that the large majority of the published literature about this topic focused on industrial countries while less than 6% focused on Asia and none on South America or Africa; they emphasize that little is known about the transferability of findings from industrial to non-industrial countries. An important input to these investigations is feedback from the population and from local aid organizations on whether alerts were received, how accurate they were (in terms of flood inundation and flood depth), how useful they were and what actions have been taken. Our research efforts are ongoing in this direction and preliminary analysis indicates flood alerts being effective. The full details of this research would be reported separately and are expected to help both in validating and in improving the flood warning system."

The reviewer is correct about the possibility of discontinuity in inundation depth due to the discretization to 16 meter pixels. This resolution was however found appropriate in providing reliable inundation prediction; the pixelated pattern better represents actually the information these maps provide. Furthermore, the feedback we received so far from governments and NGOs doesn't indicate a need for a higher resolution than that.

(j) can this study be reproduced at all? It seems not much of the study can be reproduced or even compared to in terms of data. All the code and data are either proprietary or unavailable. We were just told they could do this and do that and there is no possible path to trying most of the steps here.

We agree this is an unfavorable situation. We have no solution for the data, which we are not allowed to distribute, and the system code which is proprietary Google code. But we do want to make the models reproducible as much as we can, and in particular, the new ML inundation model presented in this paper for the first time. Therefore, we prepared a standalone Python code for the Thresholding and Manifold models, which is now on Google-research github (https://github.com/google-research/google-research/tree/master/flood_forecasting) and we cite it in the revised paper. If one has samples of gauge stage and flood inundation maps (and DEM data for the Manifold model), the code can be used for training the models and computing flood inundation and potentially their depth.

Some minor points:

Line 158. What does "State handoff" mean?

State handoff means transferring the final hidden and cell states of one LSTM, through a fully connected layer, to the initial states of the next LSTM. These are represented by the chain: h(t), c(t) -> fully connected layer - > $h_0(t),c_0(t)$ in the middle of Figure 2b. We can add this clarification in the paper if the reviewer recommends this.

Line 190. Should be "Quasi steady state" to be more exact

Thank you. We modified this term in the revised paper (Line 204).

Line 196. "Discarded" – see my point above, can you use a more gentle word?

We agree. The sentence was rephrased to (Lines 209-211):

"...it was used in the operational framework in previous seasons, but it is currently not in use since in the present conditions since it was found to be both less accurate and less scalable than the ML models (see Section 3.2)…"

Line 198-199. "when the target gauge exceeds a (pixel-specific) threshold water stage. " A bit confused. A gauge is just at one location, then why do you have a pixel-specific threshold linked to a gage? If it is pixel-specific, then you end up getting a map of different thresholds? Should it be image-specific thresholding?

The model results in a map of thresholds on the gauge stage. When the gauge data is higher than the threshold in a given pixel, this pixel is considered as wet. We do not think the term "image-specific" sufficiently clarifies this. To improve clarity we modified this text to (Lines 213-214):

"The model includes pixel-specific thresholds for each pixel in the AOI. Pixels that the water stage in the target gauge exceeds their thresholds are assumed to be inundated (i.e., wet) while the others are dry."

Line 219. Maybe I'm missing sth, although the thresholding model does not need DEM, it is tied to a particular gauge and the particular terrain/floodplain characteristics. It needs to be trained for each domain of interest using historical inundation extent and gauge height data, so it is not clear to me you can deploy to a new region without effort.

The sentence in the original paper says: "Thresholding model requires almost no site-specific data like DEMs, and no manual work, making it appealing for large scale deployment across many AOIs in a short amount of time.". We do not claim that no data at all is needed, but the data required are satellite data (i.e., SAR images for a few historical floods), which are available globally (thanks to Sentinel-1 and others). We indeed also require stage data for the target gauge, and have revised to further emphasize our focus on gauged rivers in response to previous comments. Nevertheless, we agree that it is inaccurate writing "requires almost no site-specific data" and we modified this sentence in the revised paper to (Lines 243-244):

"Thresholding model requires no DEM data but only historical flood inundation maps and gauge stages for training. This makes it appealing for large-scale deployment across many AOIs in a short amount of time."

Line 375 what happened to the flood and the effectiveness of the alert? You get us concerned but didn't say any outcome.

What we can currently say is that the alerts were sent to the public and to organizations, including CWC, which published it on Twitter (shown in Figure 7f). We were in touch with several NGOs, including the Yuganter organization, www.yuganter.org.in, that confirmed the flood events' occurrence (and sent us the photos presented in Figure 7d,e). However, we do not have yet information, which we agree is important, about actions that were taken in response to this specific flood alert. Evaluating actions on the ground requires completely distinct tools and methodologies (e.g. randomized controlled trials, field surveys, etc.). We are currently pursuing such research and hope to publish its results soon. We can share informally that preliminary results from a research collaboration with the Yale Economic Growth Center show that in places where our forecasts were distributed (through Yuganter volunteers) they have led to a statistically significant increase in protective actions compared to control cases. A discussion paragraph, presented in our response to comment i above, was added to the revised paper to better clarify this point.

**Response to reviewer #3**

We thank the reviewer for the time and effort. The comments are very helpful. Below the general comments, we copied the reviewer's comments from the paper's pdf. We present the reviewer comments in black, the responses in green, and the copied text from the manuscript in blue.

I found your manuscript very well organized and for sure impressive for the amount of information managed.

To be honest, the scientific advances you present are not enough described to make this paper useful to the community. In my opinion, this manuscript would be a perfect contribution to be published almost as is for NHESS, but fails to some part meeting the scientific innovation standard I expect in HESS.

We respectfully understand the reviewer's point here, but we argue that the revised paper would be a valid and valuable contribution to HESS. The detailed and very useful comments from this reviewer and from the two other reviewers indeed highlight aspects that required modifications such as a better description or additional information, which we addressed carefully in the revised paper, as we explain below.

We believe the scientific innovations of this study are clear in the revised paper. The most important are: 1) a full chain of flood warning systems with main core machine-learning models; 2) a new machine-learning flood inundation algorithm; 3) accuracy metrics from the "real world" of river stage and of flood inundation forecast. We believe these provide a valuable and relevant scientific contribution to HESS.

L22: How many were useful? I.e. dispatched with enough anticipation to allow for measures? How many false alarms? This is the key of success. I can also send an alert each time it rains and rivers are above a certain level at start of rainfall. Sending an alert is not a scientific measure of success

We divide our answer into two. First, regarding a scientific measure of success, note that alerts are sent based on the modeled inundation maps and river stage, both are validated against SAR inundation maps and observed stages, respectively. From cross-validation analysis, we can say something about the success or failure of our model to predict whether a given pixel is inundated or not, using the F1-score, which is computed from the elements in the contingency table, and combines hits, false alarms, and missed. Its median is 69% (Table 2, Figure 10), which is quite good. The models also predict well the water stage at the gauge locations, based on cross-validation and NSE or NSE-persist metrics (Table 2,

Figure 8). We do note claim that the number of alerts sent is a scientific measure of success, this provides an indication of scale.

The second part of the answer refers to the usefulness of the alerts. This is of course a very important issue; there is no point in sending alerts that aren't useful. Evaluating actions on the ground requires completely distinct tools and methodologies (e.g. randomized controlled trials, field surveys, etc.). We are currently pursuing such research but it was not completed yet. We can share informally that preliminary results from a research collaboration with the Yale Economic Growth Center show that in places where our forecasts were distributed (through Yuganter volunteers) they have led to a statistically significant increase in protective actions compared to control cases.

In response to this comment and another comment by reviewer #2, we added a discussion paragraph to the revised paper that explains this (Lines 535-546):

"A few studies examined the effectiveness of flood alerts in operational frameworks. For example, Rotach et al. (2009), as a part of an end-to-end operational flood warning system in the Alpine region, collected feedback from end-users through questionnaires, interviews, and workshops, and some initial insights are given on the utility of the system for the decision-makers and how well the information was perceived. It should be noted, however, that the current knowledge about effective flood warnings in countries like India and Bangladesh is very limited. For example, a literature review by Keller et al. (2021) shows that the large majority of the published literature about this topic focused on industrial countries while less than 6% focused on Asia and none on South America or Africa; they emphasize that little is known about the transferability of findings from industrial to non-industrial countries. An important input to these investigations is feedback from the population and from local aid organizations on whether alerts were received, how accurate they were (in terms of flood inundation and flood depth), how useful they were and what actions have been taken. Our research efforts are ongoing in this direction and preliminary analysis indicates flood alerts being effective. The full details of this research would be reported separately and are expected to help both in validating and in improving the flood warning system."

L49: Well, this is what we are doing for years. Most of our studies have only few years of data just because we analyze collected forecasts in operational environment:

a selection

Bogner K, Liechti K, Bernhard L, Monhart S,Zappa M. 2018. Skill of hydrological extended range forecasts for water resources management in Switzerland. Water Resources Management, 32(3), 969-984. http://doi.org/10.1007/s11269-017-1849-5

Andres N, Lieberherr G, Sideris IV, Jordan F, Zappa M. 2016. From calibration to real-time operations: an assessment of three precipitation benchmarks for a Swiss river system. Met. Apps, 23: 448–461.  HYPERLINK "http://onlinelibrary.wiley.com/doi/10.1002/met.1569/abstract"doi: 10.1002/met.1569

Liechti K, Zappa M, Fundel F, Germann U. 2013. Probabilistic evaluation of ensemble discharge nowcasts in two nested Alpine basins prone to flash floods. Hydrological processes. 27: 5-17.  HYPERLINK "http://dx.doi.org/10.1002/hyp.9458"doi:/10.1002/hyp.9458

Addor N, Jaun S, Fundel F, Zappa M. 2011. An operational hydrological ensemble prediction system for the city of Zurich (Switzerland): skill, case studies and scenarios, Hydrol. Earth Syst. Sci., 15, 2327-2347, doi:10.5194/hess-15-2327-2011. [ HYPERLINK "http://www.hydrol-earth-syst-sci.net/15/2327/2011/hess-15-2327-2011.html"Direct Link]

Zappa M, Jaun S, Germann U, Walser A, Fundel F. 2011. Superposition of three sources of uncertainties in operational flood forecasting chains. Atmospheric Research. . Thematic Issue on COST731. Volume 100, Issues 2-3, 246-262.  HYPERLINK "http://dx.doi.org/10.1016/j.atmosres.2010.12.005"doi:10.1016/j.atmosres.2010.12.005

Zappa M, Rotach MW, Arpagaus M, Dorninger M, Hegg C, Montani A, Ranzi R, Ament F, Germann U, Grossi G, Jaun S, Rossa A, Vogt S, Walser A, Wehrhan J, Wunram C. 2008. MAP D-PHASE: Real-time demonstration of hydrological ensemble prediction systems. Atmospheric Science Letters. DOI: 10.1002/asl.183

Thank you very much for those valuable references. Some of these works are relevant for the present study and we refered to those papers in the revised paper. The most relevant for our case are those presenting the full chain of tasks and models in operational frameworks and especially where evaluation metrics are shown (e.g., Zappa et al., 2008 and Addor et al., 2011).

L84-84: Nice!

Thank you.

Section 2 title: About end-to-end systems

Rotach MW, Ambrosetti P, Ament F, Appenzeller C, Arpagaus M, Bauer HS, Behrendt A, Bouttier F, Buzzi A, Corrazza M, Davolio S, Denhard M, Dorninger M, Fontannaz L, Frick J, Fundel F, Germann U, Gorgas T, Hegg C, Hering A, Keil C, Liniger MA, Marsigli C, McTaggart-Cowan R, Montani A, Mylne K, Ranzi R, Richard E, Rossa A, Santos-Muñoz D,

Schär C, Seity Y, Staudinger M, Stoll M, Volkert H, Walser A, Wang Y, Wulfmeyer V, Zappa M. 2009. MAP D-PHASE: Real-time Demonstration of Weather Forecast Quality in the Alpine Region. Bulletin of the American Meteorological Society. 90. Pages 1321-1336. HYPERLINK "http://dx.doi.org/10.1175/2009BAMS2776.1"doi:10.1175/2009BAMS2776.1

Thank you. This is indeed a relevant paper as an end-to-end operational flood warning system; it also contains an interesting part about end-user feedback. We refered to this work in the revised paper (Lines 39, 49, 536-538).

L117-121: Automatically or by operators using some kind of tools?

Automatically, as a part of the data management module. To clarify we rephrased this sentence to be (Line 117):

"Therefore, all near-real-time stage data go through a series of automatic validation and correction procedures…"

L136: And with respect to more physically-oriented models?

Both conceptual and physically-based models were compared to ML hydrological models. The sentence was modified to (Lines 138-140):

"whereas the LSTM has been shown in recent years to improve hydrological simulations relative to conceptual and physically-based models (e.g., Kratzert et al., 2018; Kratzert et al., 2019a,b; Hu et al., 2019; Feng et al., 2020; Xiang et al., 2020)."

L143: per gauge

Correct. The sentence starts with "for example, a target gauge with…" so we think this is clear, but we modified the sentence to be (Lines 146-147):

"(for example, a target gauge with a selected maximal lead time of 24 hours and hourly resolution implies 24 trained Linear models for this gauge)"

L152: Arbitrary number of steps or kind of based on analysis?

The number of steps is not arbitrary but is also not gauge-specific. We selected a number of steps to look back which we found to work well. For the system implemented in India and Bangladesh, we used 72 hours, as indicated in Table 1.

L159-161: Nice

Thank you.

L166: Described in Klotz et al., I suppose

Indeed. We added this citation again where CMAL is mentioned (Line XX).

L174-176: Reference on your validation efforts would be desirable

We are not sure what the reviewer means in this comment. The validation results are presented later on in Section 3 of the paper. We would be happy to provide more information once we understand this comment better.

L198: Well, if this is novel, it might need more detail to be reproducible. If this is a previously developed method, it needs to be referenced.

The reviewer is right, thanks for noting this. This is not a new model, but a modified version of the model presented in Ben-Haim et al., 2019. The main improvement from the original is the optimization algorithm which is described in the paper. However, we recognize that the description of the Thresholding algorithm needs more details to be better understood. The description of its categorized output into three levels of certainty was also missing. We therefore modified the text of Section 2.2a to the text presented below (Lines 212-244). Also, we prepared a standalone Python code of the inundation models which is now on Google-research github (https://github.com/google-research/google-research/tree/master/flood_forecasting), as explained below (response to comment L221). That should help in understanding the algorithm in a more complete way and make it reproducible.

"(a) Thresholding model (modified from Ben-Haim et al., 2019) (Figure 3a, a standalone code can be found in Google-research flood_forecasting, 2022): The model includes pixel-specific thresholds for each pixel in the AOI. Pixels that the water stage in the target gauge exceeds their thresholds are assumed to be inundated (i.e., wet) while the others are

dry.These thresholds are learned from the series of historic stage data at the target gauge and the corresponding state of the pixel (dry/wet) during these events (Figure 3a). Each pixel in the inundation map is treated as a separate classification task, predicting whether the pixel will be inundated or not. We refer to the "wet" class as the positive class.

The algorithm described below identifies pixel-specific thresholds and is aimed at maximizing an $F_\beta$-score using an optimized global parameter called minimal ratio. $F_\beta$-score is defined as $(1 + \beta^2) \cdot \frac{precision \cdot recall}{\beta^2 \cdot precision + recall}$ (Sokolova et al., 2006), where precision represents here the fraction of all wet pixels that are predicted as being wet and recall is the fraction of all pixels that are predicted to be wet and are really wet (see two examples for $F_1$-score in Figure S2). An iterative process is applied to each pixel. In each iteration, we find the threshold that maximizes the ratio of true wet events (where the water stage at the gauge is above the threshold and the pixel was wet) to false wet events (where the water stage at the gauge is above the threshold and the pixel is dry). The threshold that maximizes this ratio is the most cost-effective threshold in the sense that it provides the most true wet pixels per false wet instance. At the first iteration all training events are considered; then, after each selection of a threshold and its respective true-false ratio, events with stage measurements above the threshold are discarded and a new iteration starts with the remaining events. If the new true-false ratio calculated is lower than the minimal ratio parameter value, the process stops and the final threshold for the pixel is the one found in the previous iteration. It can be shown that for every minimal ratio parameter value, no other set of pixel-specific thresholds achieves simultaneously better precision and recall; implying it is Pareto optimal. Therefore, for any $\beta$ there exists some value of the minimal ratio parameter which finds the thresholds that optimize this respective $F_\beta$-score. After repeating the algorithms for different values of the minimal ratio parameter, the one that maximizes a specific target metric is selected and the respective pixel thresholds for this parameter value are used.

The Thresholding algorithm is applied to compute categories of certainty that a given pixel is wet (see an example in Figure 4c). This is done by computing the respective thresholds for two $\beta$ values of 0.3 and 3, where lower $\beta$ values imply higher thresholds. When a given water stage in the target gauge is considered, the two thresholds are examined for each pixel. If the threshold of $\beta$=0.3 is exceeded (implying the threshold of $\beta$=3 is also exceeded) it is classified as wet with a high probability. If only the thresholds of $\beta$=3 is exceeded, it is classified as wet with a low probability. The pixel is classified as dry if no threshold is exceeded.

In cases where the river stage input is higher than all past stage data, the Thresholding model's output inundation map is initialized from the most severe inundation extent seen in the historical events and expanded in all directions. The expansion distance is a linear function of the difference between the forecasted stage and the stage of the highest historical event. This Thresholding model requires no DEM data but only historical flood

L203: bit "sloppy" formulation

The reviewer is right. The text was rephrased, as presented in response to the comment above.

L221: All nice, well explained in plain text, but possibly to few information for something you declared being presented for the first time

We were debating whether to provide full details at the cost of making the paper very long and somewhat technical, vs. keeping a clear and concise explanation, albeit probably not enough to make it reproducible. Following the reviewers comments (reviewer #3 and reviewer #2) we prepared a standalone Python code for the two ML inundation models which is practically equivalent to the model runs in the operational system (now available in Google-research github https://github.com/google-research/google-research/tree/master/flood_forecasting). A citation to this code was added in the revised paper. Given this, we think it is better to leave the explanations in the main text of the paper at the current level of detail, where from the code one can understand the exact algorithm.

L271: Numbers?

We assume the reviewer is asking about the number of flood events used for training and validation. The information is given in Section 2.5 which describes the implementation of the system in India and Bangladesh. The Thresholding and Manifold models were applied to 228 gauges total; they were trained based on historical flood events for 2016-2020, where the mean number of events per AOI is 29 and the median is 15 (these numbers were modified from the those presented in the original manuscript which included the AOIs with no inundation model). We wrote the modified numbers in the revised paper.

L305: Are there also channels that do not need smartphones

Yes. The alerts are sent to relevant agencies, such as CWC, NDMA, IFRC and the Indian Red Cross, that further distribute them in their own channels. Alerts are also shown in Google Search and on Google Maps when viewing an area with an active flood alert; both

can be reached not only from smartphones but through any computer or other devices with an internet connection (see paper's Sections 2.4 and 2.5).

L326-327: Reference? Link? Documentation?

The reference is World Bank (2015). It was mentioned in the sentence that followed this text but for clarity, we added this reference also here.

L335-337: Are these separately displayed in Figure ?

No. We decided not to make different marks for different types of gauges since the map is already quite busy. But the Supplementary Table (S1) includes all the required information.

L356-366: Are you aware of action taken to mitigate loss of lives and infrastructure during this event?

This question is well justified and we are very much aware of its importance, but, unfortunately, we cannot answer it yet. We are putting a lot of effort into getting feedback from people in the area trying to get surveys either directly or through collaborations with local organizations. Unfortunately, the data we collected so far is too little and too scattered, probably, among other reasons, due to COVID limitations. So we cannot write yet about the actions that were taken. As explained in response to the comment above (L22) we have some preliminary results from our collaboration with the Yale Economic Growth Center. They show that in places where our forecasts were distributed (through Yuganter volunteers) they have led to a statistically significant increase in protective actions compared to control cases. We do hope to publish these results soon.

L358-359: Missing information on training

We are not sure what the reviewer means. The line for which this comment was written provides the mean and the median number of flood events used to train the inundation models of 2021 in India and Bangladesh. What information is missing?

L363: Wo decides?

Alerts notifications to smartphones were sent if the forecasted water stage at a gauge exceeded a pre-defined threshold, which was set by the relevant agency (CWC or BWDB). To clarify we modified this text to be (Lines 394-395):

"In cases where flooding was sufficiently severe (i.e., the forecasted water stage at the gauge exceeded a pre-defined threshold provided by CWC and BWDB), alerts were also sent as push notifications to smartphones …"

Figure 6: Really cool

Thank you.

L390-396: This is deterministic evaluation. State of the art models are probabilistic. How you plan to go for such approach?

Thanks for raising this point. First, we need to clarify that our system does include probabilistic prediction of the forecasted river stage. It is different from the standard approach to probabilistic forecasts. We first explain why it is different, and then describe the form of probabilistic forecast in our system.

Our flood warning system relies mostly on past river stage data, as opposed to typical flood warning systems where precipitation and other meteorological forcing data are the main input. Probabilistic forecast in the latter situation is very important due to: 1) high uncertainty in observed or forecasted weather and mostly in precipitation data, and, 2) high sensitivity of floods to the precipitation input. Therefore, ensembles of weather forecasts are often utilized to produce the probabilistic flood forecast. Although precipitation improves prediction accuracy to some level, in our system, at least in its current focus (i.e., gauged, large rivers), the effect of its uncertainty on the predicted flood is substantially lower (see Table 2 and Figure 8; the median NSE is increased from 0.986 to 0.987 and the median persistent-NSE from 0.66 to 0.67).

While the contribution of precipitation to uncertainty is negligible in our system, other uncertainty sources exist, with the main one being the past river stage data. The estimation of uncertainty in the forecasted river stages, resulting from past stage data and other sources, is done by estimating the time-dependent parameters of the CMAL distribution (described in Section 2.2b of the paper). The figure below (now included also in the revised paper's supplementary) demonstrates the estimated uncertainty for one gauge:

[Figure]

**Figure S1.** An example of a forecast stage graph with a 36 hour lead time (dark blue line) with the uncertainty shown as the confidence interval between the 20% and 80% quantiles (light blue shading) and the observed gauge stage (black line). Dash orange line represents the warning level above which an alert will be issued.

The real-time estimated river stage probabilities are then used in the distributed flood alerts in the form of a range of river level change (taken between the 20th and 80th quantiles). An example of such a range can be seen in Figures 4a and 4e in the paper. Furthermore, when the estimated uncertainty range exceeds a given threshold (typically 50 cm) a shorter lead time is selected for the gauge. We are currently not using the uncertainty of the stage forecasting model to inform our inundation models.

We realize that this is an important point that has to be better explained. Therefore, in the revised paper we explained better the river stage uncertainty (Lines 170-183):

"The system estimates the uncertainty of the water stage following the approach detailed in Klotz et al. (2021). The time-dependent distribution over the predicted stage is modeled using a (countable) mixture of asymmetric Laplacians (CMAL, Klotz et al., 2021). The parameters of this distribution are generated by feeding the hidden state of the forecast LSTM into a dedicated head layer for each forecasted time step. At each time step, the loss is calculated as the negative log-likelihood of the observed stages given the forecasted distribution parameters, thus maximizing the probability of the training data given model output. . It is important to note that the CMAL head is applied only to the outputs of the forecast LSTM, and the likelihood-based loss function is calculated only over these CMAL head outputs. The hindcast LSTM does not supply outputs to the loss function directly – it only affects the loss value by supplying an initial cell state to the forecast LSTM. Since training is shared for all target gauges, the maximal lead time in the training phase is taken as the maximum of the gauge-specific maximal lead times. The predicted parameters are used to calculate the distribution quantiles in each time point. We output the confidence interval between the 20% and 80% quantiles as the stage uncertainty (Figure S1) and present this range to users (see examples in Figure 4). Furthermore, when the estimated uncertainty range exceeds a given threshold (typically 50 cm), a shorter lead time is

selected for the gauge. The selected lead time is the maximal lead time such that the uncertainty range is smaller than the threshold."

We also explained the Thresholding model's support for uncertainty (Lines 234-239):

"The Thresholding algorithm is applied to compute categories of certainty that a given pixel is wet (see an example in Figure 4c). This is done by computing the respective thresholds for two $\beta$ values of 0.3 and 3, where lower $\beta$ values imply higher thresholds. When a given water stage in the target gauge is considered, the two thresholds are examined for each pixel. If the threshold of $\beta$=0.3 is exceeded (implying the threshold of $\beta$=3 is also exceeded) it is classified as wet with a high probability. If only the thresholds of $\beta$=3 is exceeded, it is classified as wet with a low probability. The pixel is classified as dry if no threshold is exceeded."

L419-420: Thank you

Thanks.

L441-442: Reference

We added a reference to:

Sokolova, M., Japkowicz, N. and Szpakowicz, S., 2006, December. Beyond accuracy, F-score and ROC: a family of discriminant measures for performance evaluation. In Australasian joint conference on artificial intelligence (pp. 1015-1021). Springer, Berlin, Heidelberg.

L494: Very nice paper and nice that you address it!

We agree this is a nice and very interesting paper.

Conclusions: Could you give some advice to teams that are not able to access to the Google infrastructure?

Thank you for this comment. It is of course very important to us that this paper would be helpful for teams working on flood forecasting, regardless of their infrastructure. After carefully considering the lesson learned from this paper and the conclusions presented, we

do not think these are constrainted by having access to Google's infrastructure. Most importantly, the stage forecast and inundation ML models presented in this paper can be implemented with standard personal computers. We want to emphasize that with this revised version we provide a stand alone Python code for the two inundation models (available at Google research github https://github.com/google-research/google-research/tree/master/flood_forecasting) that is equivalent to the code used in our system. The Linear and LSTM stage forecasting models are based on extensively published algorithms (see https://joss.theoj.org/papers/10.21105/joss.04050 for LSTM Python library). From this paper one can get benchmarks of performance levels for the ML models at least for large gauged rivers. Therefore, we believe the paper is relevant for the entire flood forecasting scientific community.

---

## Author Response (AR2)

Thank you for the paper acceptance, there were no review points to response to